# Multi-Scale Wavelet Transformers for Operator Learning of Dynamical Systems

Xuesong Wang [1]   Michael Groom [1]   Rafael Oliveira [1]   He Zhao [1]   Terence O'Kane [1]   Edwin V. Bonilla [1]

## Abstract

Recent years have seen a surge in data-driven surrogates for dynamical systems that can be orders of magnitude faster than numerical solvers. However, many machine learning-based models such as neural operators exhibit spectral bias, attenuating high-frequency components that often encode small-scale structure. This limitation is particularly damaging in applications such as weather forecasting, where misrepresented high frequencies can induce long-horizon instability. To address this issue, we propose multi-scale wavelet transformers (MSWTs), which learn system dynamics in a tokenized wavelet domain. The wavelet transform explicitly separates low- and high-frequency content across scales. MSWTs leverage a wavelet-preserving downsampling scheme that retains high-frequency features and employ wavelet-based attention to capture dependencies across scales and frequency bands. Experiments on chaotic dynamical systems show substantial error reductions and improved long-horizon spectral fidelity. On the ERA5 climate reanalysis, MSWTs further reduce climatological bias, demonstrating their effectiveness in a real-world forecasting setting.

## 1. Introduction

High-fidelity numerical solvers remain the primary tools for modeling dynamical systems governed by Partial Differential Equations (PDEs), but can be prohibitively expensive in many-query workloads (e.g., design loops and uncertainty quantification), motivating data-driven surrogates that learn simulator input-output maps and enable orders-of-magnitude faster inference once trained (Li et al., 2021; Kovachki et al., 2023). Among these, neural operators learn a map between function spaces that can be discretization-

invariant, which allows training on one discretization and evaluation on others, as exemplified by DeepONet and the Fourier Neural Operator (Lu et al., 2021; Li et al., 2021; Oliveira et al., 2025). However, these advantages do not come without limitations. A key failure mode in neural operators is spectral bias, sometimes referred as oversmoothing, due to the tendency of neural networks to learn low-frequency components faster and more reliably than high-frequency components (Rahaman et al., 2019; Cao et al., 2021; Wang et al., 2021). In the context of neural operators, spectral bias manifests as systematic under/overestimation of high-wavenumber content, which is particularly harmful for multi-scale and chaotic systems where small-scale errors can rapidly degrade large-scale predictions (Chattopadhyay et al., 2023). The cause of spectral bias can be mainly attributed to two aspects: architectural bottlenecks that limit high-frequency feature representation such as Fourier-mode truncation (Koshizuka et al., 2024); and standard training objectives such as MSE-type losses that can overweight low-frequency energy and encourage smooth solutions (Subich et al., 2025).

Existing work addresses spectral bias in neural operators through three primary directions. **(i) Multi-scale architectures** (Rahman et al., 2022; Liu et al., 2024; Raonic et al., 2023) propagate information across resolutions and recover fine-scale structures. While effective in many regimes, multi-scale feature extraction alone does not guarantee that high-frequency content is preserved when downsampling is aggressive. **(ii) Frequency-aware representations** (Khodakarami et al., 2025; Meyer, 1992; Gupta et al., 2021) explicitly separate low/high-frequency components via heuristics or wavelet transforms. These methods often provide only coarse frequency separation, and convolutional instantiations (Tripura & Chakraborty, 2023) sacrifice expressiveness for tractability. Recent transformer-based advances (Zhou et al., 2026) exploit attention for modeling wavelet feature dependencies but are directly applied on the physical grid which can be computationally expensive for high-resolutions. **(iii) Objective functions and rollout regularization** (Liu et al., 2024; Hu et al., 2025; Lippe et al., 2023; Li et al., 2022; He et al., 2025) alter training objectives to penalize small-scale errors or improve long-term stability. While largely complementary to architectural choices, diffusion-based members of this family introduce

[1]CSIRO, Australia. Correspondence to: Edwin V. Bonilla <edwin.bonilla@csiro.au>.

*Proceedings of the 43rd International Conference on Machine Learning*, Seoul, South Korea. PMLR 306, 2026. Copyright 2026 by the author(s).

additional inference cost through multi-step sampling.

In this work, we focus on proposing an architecture that enhances the frequency-aware representation in a multi-scale manner, trained with the standard relative $\ell_2$ (MSE-type) training objective, to isolate the role of representation and inductive bias. The resulting method, **Multi-Scale Wavelet Transformer (MSWT)**, operates in a wavelet space underlying a physical dynamical system. The design has three components: (i) a lightweight patch tokenizer mapping the physical field to tokens, (ii) a multi-scale wavelet down/up-sampling scheme in token space that explicitly preserves low-frequency and high-frequency sub-bands across scales; and (iii) a wavelet-based attention that models dependencies across wavelet sub-bands. Our main contributions are:

1. **Mitigating spectral bias via wavelet attention.** We introduce a wavelet-based transformer architecture that models multiband wavelet content across scales, resulting in improved spectral fidelity.

2. **Improved efficiency and long-horizon stability.** By combining tokenization with wavelet-based down/up sampling and an efficient wavelet attention mechanism, we reduce attention cost significantly while improving stability and accuracy in long rollouts of chaotic dynamical systems.

3. **Long-horizon evaluation on ERA5 with climatology metrics.** We validate on the ERA5 atmospheric reanalysis of the global climate (Hersbach et al., 2020) and demonstrate reduced climatological bias in free-running forecasts, indicating improved long-term physical consistency.

Our code is publicly available at `https://github.com/csiro-funml/MSWT`.

**Conflict of Interest Disclosure.** The authors declare no conflicts of interest. This work received no external funding.

## 2. Related Work

**Neural operators.** Neural operators have emerged as a general paradigm for surrogate modeling of PDEs and dynamical systems (Lu et al., 2021; Kovachki et al., 2023). A widely used backbone is the Fourier Neural Operator (FNO), which performs global mixing of Fourier features through truncated Fourier representations (Li et al., 2021). FNOs can also be coupled with Bayesian optimization via Thompson sampling (Oliveira et al., 2025) to efficiently find the optimal input function that maximizes a functional of an unknown operator's output, with applications to topology optimization and inverse problems governed by PDEs. More recently, transformer-style token mixing has been adapted to operator

learning, including attention-based and tokenized formulations (Cao, 2021; Hao et al., 2023; Wu et al., 2024; Li et al., 2023; Hao et al., 2024). Continuous or geometry-aware variants further extend attention-based operators beyond regular grids (Wang et al., 2025; Wu et al., 2024). These developments motivate architectures that combine the expressiveness of attention with inductive biases suited for multi-scale physics. Our MSWT follows this direction by moving attention into a wavelet space representation.

**Understanding and mitigating spectral bias.** Recent works have investigated why operator learners misrepresent fine scales and how this impacts long-horizon rollouts. Learning-dynamics analyses based on the frequency principle argue that gradient-based optimization tends to fit coarse structures first, slowing convergence of fine-scale components and inducing oversmoothing in multi-scale PDEs (Xu et al., 2025). Related perspectives such as eigenvector bias show that feature parameterizations can preferentially align with certain spectral components, making multi-scale PDEs challenging even with Fourier features (Wang et al., 2021). For neural operators, spectral studies of FNO-style models report systematic difficulty in learning non-dominant (often higher-wavenumber) modes (Qin et al., 2024), and mean-field expressivity results identify Fourier-mode truncation as an explicit bandwidth bottleneck that limits representable fine-scale content (Koshizuka et al., 2024). Mitigation strategies include multi-scale architectures that improve cross-resolution information flow (Rahman et al., 2022; Liu et al., 2024; Raonic et al., 2023) and frequency-aware reweighting or scaling heuristics that amplify high-frequency error signals (Khodakarami et al., 2025). A complementary direction learns dynamics in compressed or latent representations, where reduced-order structure can alleviate error accumulation and improve long-horizon stability and spectral behavior (Kontolati et al., 2024; Li et al., 2025). This motivates our use of a lightweight patch tokenizer to embed high-dimensional fields into a token space for attention.

**Wavelet transforms for mitigating spectral bias.** Wavelet transforms provide a principled representation that extracts global frequency signals while retaining localized spatial information (Meyer, 1992). This is particularly attractive for operator learning because high-frequency components are often localized (sharp gradients), and a wavelet basis can preserve these structures more naturally than global Fourier bases. Wavelet-based neural operators incorporate wavelet transforms into operator layers and demonstrate improved multi-scale representation compared to purely Fourier-based mixing (Gupta et al., 2021; Tripura & Chakraborty, 2023). However, many existing wavelet operators primarily rely on linear or convolutional mixing in wavelet space, which can limit expressiveness when modeling nonlinear cross-scale interactions. Recent

transformer hybrids combine attention with spectral processing (Zhou et al., 2026), but typically apply attention on the full-resolution grid, which can become computationally expensive at high resolution. In contrast, our MSWT is designed to address both limitations. First, attention is applied directly in a tokenized wavelet representation, enabling nonlinear interactions across wavelet subbands and scales. Second, wavelet-preserving downsampling allows attention to operate on a reduced-resolution domain while retaining fine-scale information, rather than discarding high-frequency structure through conventional pooling or subsampling. The localized nature of the wavelet representation also naturally supports stable boundary handling in spherical domains.

**Weather and climate emulation with long-horizon evaluation.** Data-driven emulators trained on European Centre for Medium-Range Weather Forecasts reanalysis 5 (ERA5) (Hersbach et al., 2020) have achieved strong short-to-medium range performance with spectral/operator-inspired backbones (Guibas et al., 2022; Pathak et al., 2022; Bonev et al., 2025). For free-running long-horizon simulation, however, the main focus is on distributional drift, i.e., small-scale errors accumulate into systematic bias in climatological statistics (Chattopadhyay et al., 2023). Accordingly, recent efforts emphasize stability and physical consistency over long horizons (Kochkov et al., 2024; Guan et al., 2025), motivating our focus on long-horizon rollouts evaluated with climatology metrics on ERA5.

It is worth clarifying that *climate* emulation over multi-decadal autoregressive rollouts is fundamentally distinct from short- and medium-range *weather* forecasting. Weather forecasting systems such as GraphCast (Lam et al., 2023) and FourCastNet (Bonev et al., 2025) primarily optimize predictive skill at lead times of days to weeks, over which small-scale error accumulation may not substantially impact the prediction. In contrast, long-horizon climate emulation focuses on controlling distributional drift, spectral bias, and long-term stability under repeated rollout, while operating in a comparatively small-data regime due to the limited historical climate record (relative to the timescales involved). In our ERA5 experiments, we therefore compare primarily against LUCIE (Guan et al., 2025), which shares the same long-horizon climate-emulation setting and similarly emphasizes stable autoregressive dynamics and climatological consistency over multi-decadal timescales.

## 3. Problem formulation

Let $\Omega \subseteq \mathbb{R}^d$ be a spatial domain and let $u(\boldsymbol{s}, t) \in \mathbb{R}^{C_u}$ denote the state of a dynamical system (e.g., a PDE solution field) at location $\boldsymbol{s} \in \Omega$ and time $t$. We consider learning an operator $\mathcal{G}$ for a future step prediction: given a state at time $t$, predict the state at time $t + 1$ in discrete time

intervals of length $\Delta t > 0$. Formally, we aim to learn an operator $\mathcal{G} : u_t(\cdot) \mapsto u_{t+1}(\cdot)$, where $u_t(\cdot) := u(\cdot, t)$. In practice, we observe discretized fields on a uniform grid of size $H \times W$. We write $\boldsymbol{U}_t \in \mathbb{R}^{H \times W \times C_u}$ as the discretized $u_t$ and learn a parametric approximation $\mathcal{G}_\theta$. Our training set consists of one-step pairs $\{(\boldsymbol{U}_t^{(i)}, \boldsymbol{U}_{t+1}^{(i)})\}_{i=1}^N$ extracted from trajectories with multiple initial conditions of PDEs. To provide spatial context, we augment the state with positional features $v(\boldsymbol{s}) \in \mathbb{R}^{C_v}$. Given coordinates $v(\boldsymbol{s})$, we form the model input as $x_t(\boldsymbol{s}) = \begin{bmatrix} u_t(\boldsymbol{s}), v(\boldsymbol{s}) \end{bmatrix} \in \mathbb{R}^{C_{\text{in}}}$, discretized as $\boldsymbol{X}_t \in \mathbb{R}^{H \times W \times C_{\text{in}}}$. The model outputs $\hat{\boldsymbol{U}}_{t+1} = \mathcal{G}_\theta(\boldsymbol{X}_t) \in \mathbb{R}^{H \times W \times C_u}$.

**Wavelet transforms.** Similar to the functional basis in Fourier transforms, discrete wavelet transforms (DWT) use a mother wavelet $\psi$ to represent a function across multiple spatial scales, providing localized multiresolution information (Mallat, 2008). We use the Haar wavelet to explicitly preserve fine-scale content. It yields an exact multiresolution decomposition into low-frequency and high-frequency subbands with the normalized low-pass and high-pass filters: $\boldsymbol{g} = \frac{1}{\sqrt{2}}[1, 1], \boldsymbol{h} = \frac{1}{\sqrt{2}}[1, -1]$. Here $\boldsymbol{g}$ is a low-pass (smoothing) filter because it averages adjacent samples: applying $\boldsymbol{g}$ computes $(x_0 + x_1)/\sqrt{2}$. Conversely, $\boldsymbol{h}$ is a high-pass (detail) filter because it takes a local difference: applying $\boldsymbol{h}$ computes $(x_0 - x_1)/\sqrt{2}$, which responds strongly to sharp transitions. See Appendix A for further background.

**2D Discrete Wavelet Transform** The 2D Haar DWT is separable by applying $\boldsymbol{g}$ or $\boldsymbol{h}$ along each axis and produces four subbands per channel: LL (low-low), LH (low-high), HL (high-low), and HH (high-high). Given a feature map $\boldsymbol{X} \in \mathbb{R}^{\times H \times W \times C}$, four separable 2D filters for the Haar wavelet are defined as $\boldsymbol{\Psi}_{\text{LL}} = \boldsymbol{g} \otimes \boldsymbol{g}, \boldsymbol{\Psi}_{\text{HL}} = \boldsymbol{h} \otimes \boldsymbol{g}, \boldsymbol{\Psi}_{\text{LH}} = \boldsymbol{g} \otimes \boldsymbol{h}$ and $\boldsymbol{\Psi}_{\text{HH}} = \boldsymbol{h} \otimes \boldsymbol{h}$ to obtain:

$$\boldsymbol{\Psi}_{\text{LL}} = \tfrac{1}{2}\begin{bmatrix} 1 & 1 \\ 1 & 1 \end{bmatrix}, \qquad \boldsymbol{\Psi}_{\text{LH}} = \tfrac{1}{2}\begin{bmatrix} 1 & -1 \\ 1 & -1 \end{bmatrix},$$
$$\boldsymbol{\Psi}_{\text{HL}} = \tfrac{1}{2}\begin{bmatrix} 1 & 1 \\ -1 & -1 \end{bmatrix}, \quad \boldsymbol{\Psi}_{\text{HH}} = \tfrac{1}{2}\begin{bmatrix} 1 & -1 \\ -1 & 1 \end{bmatrix}, \quad (1)$$

where $\otimes$ denotes the outer product. A 2D DWT step is given by convolutions with stride 2:

$$\mathcal{W}(\boldsymbol{X}) = \begin{bmatrix} \boldsymbol{X} * \boldsymbol{\Psi}_{\text{LL}}, \ \boldsymbol{X} * \boldsymbol{\Psi}_{\text{LH}}, \ \boldsymbol{X} * \boldsymbol{\Psi}_{\text{HL}}, \ \boldsymbol{X} * \boldsymbol{\Psi}_{\text{HH}} \end{bmatrix}, (2)$$

where $*$ denotes convolution, and the output is concatenated along the channel dimension. Thus $\hat{\boldsymbol{X}} = \mathcal{W}(\boldsymbol{X}) \in \mathbb{R}^{\frac{H}{2} \times \frac{W}{2} \times 4C}$. The inverse transform (iDWT) $\mathcal{W}^{-1}$ reconstructs $\boldsymbol{X}$ from its four subbands via upsampling and transposed depthwise convolutions with the exact filters, therefore $\mathcal{W}^{-1}(\hat{\boldsymbol{X}}) \in \mathbb{R}^{H \times W \times C}$.

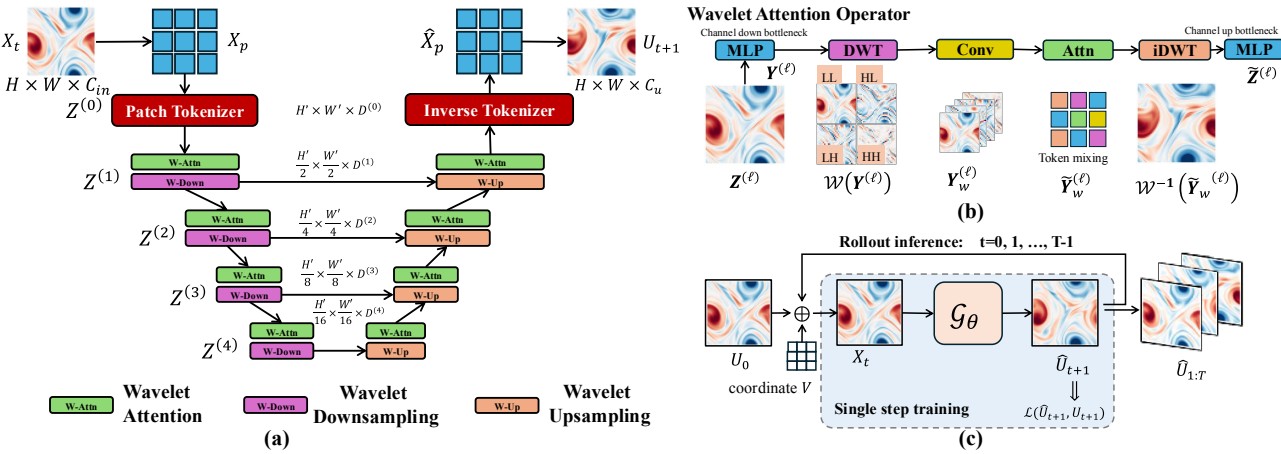

*Figure 1*. Overview framework of Multi-Scale Wavelet Transformer (MSWT). Figure 1(a) presents the U-shaped topology. Inputs are embedded into patch tokens, processed by wavelet-attention blocks at multiple scales, and down/up-sampled via Wavelet-based sampling. Figure 1(b) illustrates the mechanism of the wavelet attention operator where the token mixing attention is achieved in the wavelet space. $\mathcal{W}$ represents the discrete wavelet transform (DWT) extracting features of four frequency bands (low-low, low-high, high-low, high-high) and $\mathcal{W}^{-1}$ represents the inverse transform (iDWT) to recover the token space. Figure 1(c) shows the training and inference of MSWT for operator learning of dynamical systems. $\mathcal{G}_\theta$ includes all the parameters $\theta$ in the operator $\mathcal{G}$.

## 4. Multi-Scale Wavelet Transformer (MSWT)

We propose MSWT, an operator learner with frequency-enhanced representations in a multi-scale manner that mitigates spectral bias.

**Motivation and design principle.** Long-horizon autoregressive prediction in multi-scale dynamical systems is highly sensitive to the treatment of fine-scale structure. Small errors at high spatial frequencies do not remain localized: under nonlinear dynamics, they propagate across scales and accumulate over a rollout, eventually degrading large-scale coherence and long-term stability. Standard downsampling or pooling operations can exacerbate this effect by discarding high-frequency information in intermediate representations. MSWT is therefore designed to address this problem through a multiresolution wavelet representation that preserves both coarse and fine scales throughout the network hierarchy. Rather than applying attention directly on the full-resolution grid, we first transform the state into a tokenized wavelet representation and perform attention in reduced-resolution wavelet spaces. This enables efficient modeling of long-range interactions while retaining localized high-frequency structure and cross-scale dependencies that are important for stable long-horizon prediction.

**Architecture overview.** As shown in the overview in Fig. 1(a), MSWT has two main components: a patch tokenizer/inverse tokenizer that learns the mapping between the physical field and a token space, where the spatially local features (usually high frequency) are aggregated; (b) a multi-scale structure that sequentially extracts frequency features from fine scales to coarse scales. Inside each scale

we have a wavelet attention block that models the dependencies across frequency bands and a wavelet-preserving down-/up-sampling block that models the change of effective resolution without oversmoothing. In the following, we introduce each component.

### 4.1. Patch Tokenizer: Physical grid to Tokens

Our patch tokenizer maps the physical field to the token space so that the resulting loss landscape becomes smoother to optimize. Let $\boldsymbol{X}_t \in \mathbb{R}^{H \times W \times C_{\text{in}}}$ be the input field. We partition the grid into non-overlapping patches of size $p \times p$ and map each patch to a token. Let $\Pi_p(\cdot)$ denote patchification, which reshapes the input into $\boldsymbol{X}_p = \Pi_p(\boldsymbol{X}_t) \in \mathbb{R}^{H_0 \times W_0 \times (p^2 C_{\text{in}})}$, where $H_0 = \frac{H}{p}, W_0 = \frac{W}{p}$. We then apply a mapping $\phi_{\text{in}}$ to obtain the initial token representation $\boldsymbol{Z}^{(0)} = \phi_{\text{in}}(\boldsymbol{X}_p) \in \mathbb{R}^{H_0 W_0 \times D_0}$, where $D_0$ is the token dimension. We found tokenization a crucial step to reduce the effective resolution for expensive attention operations and improve computational efficiency for high-resolution PDEs. The mapping from physical space to token space improves stability for long-horizon rollouts, which makes the following multi-scale structure operate on a smoother optimization landscape and in turn reduce the model's spectral bias.

For cases where our targets consist of periodic PDE data, we implement circular padding for periodic wavelet transforms. When $H$ or $W$ is odd, we pad the bottom/right boundary via a circular wrap, which ensures that down/up-sampling respects periodic boundary conditions and avoids artifacts induced by zero padding.

## 4.2. Multi-scale Structure

Our multi-scale structure obtains several scales of frequency-aware features for operator learning, representing the ground truth in both small-scale details (high-frequency components) and large scales (low-frequency components). As shown in Fig 1(a), MSWT uses $L$ scales of downsampling and upsampling regimes. At scale $\ell \in \{0, \ldots, L-1\}$, we maintain a token sequence $\boldsymbol{Z}^{(\ell)} \in \mathbb{R}^{N_\ell \times D_\ell}, N_\ell = H_\ell W_\ell$, with spatial resolution $(H_\ell, W_\ell)$ and channel dimension $D_\ell$. The encoder stages iteratively apply a wavelet attention block (WAttn) followed by wavelet down-sampling ($\downarrow$), and the decoder symmetrically applies wavelet up-sampling ($\uparrow$) followed by an attention block:

$$\boldsymbol{Z}^{(\ell)} \xrightarrow{\text{WAttn}} \tilde{\boldsymbol{Z}}^{(\ell)} \xrightarrow{\downarrow} \boldsymbol{Z}^{(\ell+1)}, \quad (3)$$

$$\boldsymbol{Z}^{(\ell+1)} \xrightarrow{\uparrow} \hat{\boldsymbol{Z}}^{(\ell)} \xrightarrow{\text{WAttn}} \boldsymbol{Z}^{(\ell)}. \quad (4)$$

We now describe each block, namely wavelet attention block and wavelet-preserving down-/ up-sampling.

### 4.2.1. WAVELET ATTENTION BLOCK

A wavelet attention block aims to model the token mixing in the frequency space such that different frequency subbands can be aggregated without spectral misrepresentation. The overview of wavelet attention follows a transformer architecture, i.e., given a scale-specific $\boldsymbol{Z}$ (we suppress $\ell$ for clarity), the wavelet attention block WAttn($\boldsymbol{Z}$) computes

$$\boldsymbol{Z} \leftarrow \boldsymbol{Z} + \text{WAO}(\text{LN}(\boldsymbol{Z})), \quad \tilde{\boldsymbol{Z}} \leftarrow \boldsymbol{Z} + \text{FFN}(\text{LN}(\boldsymbol{Z})), \quad (5)$$

where LN is layer normalization and FFN is a feed forward network and (WAO) stands for the **wavelet attention operator**. As detailed in Fig 1 (b), WAO consists of: (i) two projection layers (MLP) (front and end) to compress and recover the input dimension ($D \leftrightarrow D/4$) for a parameter-efficient wavelet transform; (ii) DWT to extract the wavelet coefficients and iDWT to recover them to the token space; and (iii) a convolution followed by attention to model both local and global dependencies of different frequency subbands. We will now provide more details on the modelling in the wavelet space, assuming $\boldsymbol{Y} \in \mathbb{R}^{H \times W \times \frac{D}{4}}$ is the representation of $\boldsymbol{Z}$ after the dimension compression.

**DWT to obtain low and high frequency efficiently.** We apply the 2D DWT with the Haar wavelets on $\boldsymbol{Y}$ to obtain wavelet coefficients

$$\boldsymbol{Y}_w = \mathcal{W}(\boldsymbol{Y}) \in \mathbb{R}^{H' \times W' \times D}. \quad (6)$$

The wavelet space $H' = \frac{H}{2}, W' = \frac{W}{2}$ is halved in the spatial dimension and the channel features are stacked together to include all four components of the DWT, which represent both the "smoothed" version and several levels of "details" in the token space (see Fig 1 (b) DWT for a demonstration

of four components of a PDE input). We then introduce the modeling layer using WAO to learn the dependencies among various frequency subbands.

**Convolution and wavelet-attention to model frequency dependencies in the wavelet space**. To enhance the modeling capacity of mixing both the local and global wavelet coefficients, we first use a convolution to mix the *local wavelet information* and then compute the self-attention to learn *global wavelet mixing*:

$$\tilde{\boldsymbol{Y}}_w = \text{Attn}\Big(\text{Conv}(\boldsymbol{Y}_w)\Big) = \text{softmax}\left(\frac{\boldsymbol{Q}\boldsymbol{K}^\top}{\sqrt{D}}\right)\boldsymbol{V}, \quad (7)$$

with learned projections $\boldsymbol{Q} = \text{Conv}(\boldsymbol{Y}_w)\boldsymbol{W}_Q$, $\boldsymbol{K} = \text{Conv}(\boldsymbol{Y}_w)\boldsymbol{W}_K$, $\boldsymbol{V} = \text{Conv}(\boldsymbol{Y}_w)\boldsymbol{W}_V$, $N' = H'W'$ and channel dimension $D$. Here the convolution and the attention mix the features across different frequency bands, enabling an expressive model to amplify high frequency components usually with small energy. We further exploit the local window attention strategy (Liu et al., 2021) to only compute attention within a window of size $m$, which significantly reduces the complexity from $\mathcal{O}(N'^2)$ to $\mathcal{O}(N'm^2)$ given that $N' \gg m^2$.

**iDWT to map the wavelet coefficients back to the token space**. We will then map the wavelet coefficients back to the token space by applying the inverse transform and recover the wavelet resolution as well:

$$\tilde{\boldsymbol{Y}} = \mathcal{W}^{-1}(\tilde{\boldsymbol{Y}}_w) \in \mathbb{R}^{H \times W \times \frac{D}{4}}. \quad (8)$$

Finally, we use the final projection layer to recover the channel dimension to $D$. We summarize the whole wavelet attention operator as $\tilde{\boldsymbol{Z}} = \text{WAttn}(\boldsymbol{Z})$. The overall design of the wavelet attention block has two practical benefits: (i) attention is applied at reduced spatial resolution $(H/2, W/2)$ for global window attention or $(m, m)$ using local window attention strategy; and (ii) all four wavelet subbands are explicitly present in $\boldsymbol{Y}_w$, encouraging the model to preserve and mix high-frequency content rather than discarding it via pooling, as our experiments demonstrate.

### 4.2.2. WAVELET-PRESERVING DOWN-SAMPLING

Now that we extract the wavelet features at one scale, a straightforward generalization is to gradually get more features across different scales. Unlike standard downsampling, which may discard high-frequency information (Dumitrescu & Boiangiu, 2019), our down-sampling uses the DWT to carry all subbands to the next scale. Given tokens $\tilde{\boldsymbol{Z}}^{(\ell)}$, we first apply a linear mapping $\phi_\downarrow$ to compress the channel dimension $D_\ell \to D_\ell/4$, reshape to a grid and apply DWT, and finally use a convolution to map the channels to the next scale dimension and enhance local dependency (Yao et al.,

2022; Zhou et al., 2026):

$$\boldsymbol{Z}^{(\ell+1)} = \text{Conv}\Big[\mathcal{W}\Big(\phi_\downarrow(\tilde{\boldsymbol{Z}}^{(\ell)})\Big)\Big] \in \mathbb{R}^{H_{\ell+1} \times W_{\ell+1} \times D_{\ell+1}},$$
(9)

where $H_{\ell+1} = H_\ell/2$ and $W_{\ell+1} = W_\ell/2$. After stacking several scales of downsampling layers, our model is capable of extracting wavelet features from both the fine scale (small-scale details) to coarse scale (large-scale patterns), though there is a trade-off between computational efficiency and stability versus preserving fine-scale detail: aggressive compression often induces oversmoothing, while architectures that retain high-frequency structure, such as attention, increase compute and can raise the risk of overfitting with limited data.

### 4.2.3. WAVELET-PRESERVING UP-SAMPLING

Decoder up-sampling mirrors the encoder and uses iDWT to increase resolution while preserving band structure. Given $\boldsymbol{Z}^{(\ell+1)}$, (i) we project channels via $\phi_\uparrow$; (ii) we apply iDWT to obtain an upsampled feature map $\mathcal{W}^{-1}(\hat{\boldsymbol{U}})$; and (iii) we concatenate the skip features and fuse with a convolution:

$$\hat{\boldsymbol{Z}}^{(\ell)} = \text{Conv}\Big[\big(\mathcal{W}^{-1}(\phi_\uparrow(\boldsymbol{Z}^{(\ell+1)}))\big),\ \boldsymbol{Z}^{(\ell)}\Big] \in \mathbb{R}^{H_\ell \times W_\ell \times D_\ell}.$$
(10)

As stated earlier, the high frequency components (small-scale details) are not smoothed out but reweighted, potentially amplified by the downsampling operation. Using the skipping layer and iDWT, these high-frequency features are recovered in the original token space. After each step, the effective resolution was halved in each dimension, significantly improving computational cost for wavelet attention.

### 4.3. Inverse Tokenizer, Training Objective and Autoregressive Rollout

Finally, we map the token features $\boldsymbol{Z}^{(0)} \in \mathbb{R}^{N_0 \times D_0}$ back to the physical space via the inverse tokenizer. We map the tokens back to the physical grid: $\hat{\boldsymbol{X}}_p = \phi_{\text{out}}(\boldsymbol{Z}^{(0)}) \in \mathbb{R}^{H_0 \times W_0 \times (p^2 C_u)}$ and unpatchify: $\hat{\boldsymbol{U}}_{t+1} = \Pi_p^{-1}(\hat{\boldsymbol{X}}_p) \in \mathbb{R}^{H \times W \times C_u}$. As illustrated in Fig 1(c), we train MSWT on one-step pairs using the standard relative $\ell_2$ loss:

$$\mathcal{L}(\theta) = \mathbb{E}_{(\boldsymbol{U}_t, \boldsymbol{U}_{t+1})}\left[\frac{\|\mathcal{G}_\theta(\boldsymbol{X}_t) - \boldsymbol{U}_{t+1}\|_2}{\|\boldsymbol{U}_{t+1}\|_2 + \varepsilon}\right], \quad \varepsilon > 0$$
(11)

At inference time, we obtain long-horizon rollouts by iterating the learned operator given the same spatial coordinate $\boldsymbol{V} : \hat{\boldsymbol{U}}_{t+k+1} = \mathcal{G}_\theta([\hat{\boldsymbol{U}}_{t+k}, \boldsymbol{V}]), k = 0, 1, \ldots$, with $\hat{\boldsymbol{U}}_t = \boldsymbol{U}_t$ at the initial step. This free-running setting is used in our long-horizon stability evaluations.

## 5. Experiments

**Baselines.** We compare the proposed Multi-Scale Wavelet Transformer against five competitive baselines for dynamical simulators: (a) the Fourier Neural Operator (**FNO**) (Li et al., 2021), a standard spectral neural operator for PDE surrogate modeling; (b) **U-Net** (Ronneberger et al., 2015), a strong multiresolution convolutional baseline; (c) the Wavelet Neural Operator (**WNO**) (Tripura & Chakraborty, 2023), which injects multilevel wavelet transforms and serves as a direct wavelet-based counterpart; (d) the Spectral Attention Operator Transformer (**SAOT**) (Zhou et al., 2026), which integrates spectral operators (e.g., FNO-like mixing) and wavelet structure within attention, providing a closely related transformer baseline; and (e) High-Frequency Scaling (**HFS**) (Khodakarami et al., 2025), a U-Net variant that explicitly amplifies high-frequency components to mitigate spectral bias, which is especially relevant for long-horizon rollouts. For the real-world weather benchmark, we additionally include the Lightweight Uncoupled ClImate Emulator (**LUCIE**) (Guan et al., 2025), built on the Spherical Fourier Neural Operator (Bonev et al., 2023) and designed to produce stable, physically consistent climate predictions over decadal time scales. We set the hyperparameters of the competing methods to have approximately the same parameter count as ours. We elaborate more on the hyperparameter setting and training setups in Appendix B.2.1.

**Benchmarks.** We consider two simulated fluid-dynamics benchmarks and one real-world benchmark for weather prediction. The Chaotic Kolmogorov Flow (**CKF**) (Li et al., 2024) is a 2D flow governed by the Navier-Stokes equations for a viscous, incompressible fluid. We use a $64 \times 64$ vorticity field and evaluate both short- and long-horizon rollouts. The Shallow Water Equation (**SWE**) (Gupta & Brandstetter, 2022) is a 2D flow arising from cases where the horizontal length scales are much greater than the vertical length scale, which makes it a commonly used proxy for weather-like dynamics. We model SWE on a $96 \times 192$ grid using the vorticity and pressure fields. Finally, we evaluate on **ERA5** (Hersbach et al., 2020), a global reanalysis that provides hourly estimates of atmospheric variables over multiple decades. We use five prognostic variables and two forcing variables as inputs at $48 \times 96$ resolution, and predict future prognostic variables along with one diagnostic variable, precipitation. More details about the benchmarks can be found in Appendix B.2 to Appendix B.4.

**Evaluation.** We evaluate the model in terms of its relative error norm and spectrum behaviour in short- and long-term rollouts. More specifically, for two simulation benchmarks, we choose standard relative $L^2$ norm (Li et al., 2021) which measures the norm of the prediction error normalized by the ground truth norm. To further showcase how well the

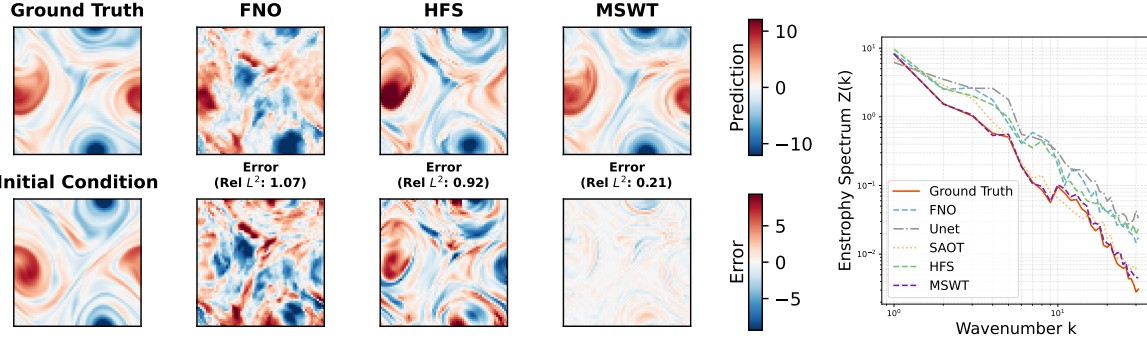

*(a)* Rollout predictions and error comparison at $t = 64$.       *(b)* Enstrophy power spectrum.

*Figure 2.* Predictions and enstrophy power spectrum on the CKF dataset for long-term rollouts. We present the 64-step rollout predictions of the baselines and our approach. More comparisons can be found in the Appendix B.2.2.

*Table 1.* CKF short-/long-term evaluation (mean $\pm$ std) for **Rel** $L^2$, **EMLR**, and **EMAE** ($\downarrow$). Best entries are highlighted. "–" indicates unstable long rollouts. The last row reports the relative improvement of MSWT over the second-best stable model.

| Model | Rel $L^2$ step 1 | step 30 | step 64 | EMLR step 1 | step 30 | step 64 | EMAE step 1 | step 30 | step 64 |
|---|---|---|---|---|---|---|---|---|---|
| FNO | $0.05 \pm 0.00$ | $0.57 \pm 0.02$ | $0.91 \pm 0.02$ | $0.08 \pm 0.00$ | $0.52 \pm 0.01$ | $0.63 \pm 0.02$ | $0.07 \pm 0.00$ | $0.83 \pm 0.02$ | $1.08 \pm 0.06$ |
| Unet | $0.10 \pm 0.09$ | – | – | $0.06 \pm 0.06$ | $1.41 \pm 0.27$ | $6.08 \pm 1.72$ | $0.86 \pm 1.85$ | – | – |
| WNO | $0.25 \pm 0.00$ | – | – | $0.37 \pm 0.00$ | – | – | $5.20 \pm 0.00$ | – | – |
| SAOT | $0.06 \pm 0.00$ | $0.86 \pm 0.10$ | $1.25 \pm 0.28$ | $0.04 \pm 0.01$ | $0.53 \pm 0.25$ | $0.92 \pm 0.50$ | $0.04 \pm 0.01$ | $0.87 \pm 0.60$ | $1.98 \pm 1.77$ |
| HFS | $0.05 \pm 0.00$ | $0.70 \pm 0.01$ | $0.94 \pm 0.01$ | $0.04 \pm 0.00$ | $0.58 \pm 0.05$ | $0.85 \pm 0.07$ | $0.04 \pm 0.00$ | $1.02 \pm 0.14$ | $1.95 \pm 0.34$ |
| MSWT | $\mathbf{0.02 \pm 0.00}$ | $\mathbf{0.20 \pm 0.02}$ | $\mathbf{0.33 \pm 0.03}$ | $\mathbf{0.03 \pm 0.00}$ | $\mathbf{0.16 \pm 0.02}$ | $\mathbf{0.20 \pm 0.02}$ | $\mathbf{0.02 \pm 0.00}$ | $\mathbf{0.15 \pm 0.01}$ | $\mathbf{0.21 \pm 0.02}$ |
| Rel. improv. (%) | 60.0 | 64.9 | 63.7 | 25.0 | 69.8 | 76.5 | 50.0 | 81.9 | 80.6 |

model matches the small details that contain relatively lower energy, we propose to use four additional metrics: spectral / enstrophy mean absolute error (SMAE/ EMAE), spectral / enstrophy mean log ratio (SMLR/ EMLR) based on the power spectra for the two invariant quantities: enstrophy (which represents the "intensity" of the vorticity) and kinetic energy. All five metrics are evaluated at different time steps. For detailed computation of the metrics, see Appendix B.1.

### 5.1. Chaotic Kolmogorov Flow (CKF)

In this simulation benchmark, we evaluate the model's ability to track fine-scale details that evolve chaotically. We report the Relative $L^2$ norm, EMLR, and EMAE of the compared methods at three time steps (1, 30, and 64) in Table 1 (additional SMLR and SMAE results are provided in Appendix Table 5). We also visualize predictions of selected methods at step 64 in Fig. 2a (more predictions and energy-spectrum results for steps 1 and 30 are included in Appendix Fig. 5 through Appendix Fig. 10). MSWT consistently outperformed the baselines across time steps, achieving the lowest errors and the slowest growth rate. In particular, MSWT improved upon the second-best model (FNO) by over 60% in Relative $L^2$. Unet and WNO became unstable quickly and failed by step 30. Other wavelet-based state-of-the-art methods, including HFS and SAOT, exhibited substantially stronger error accumulation than MSWT.

We attribute this advantage to representing physical fields in a token space where the dynamics are smoother and more stable for long-horizon prediction.

The performance gap is also evident in EMLR and EMAE, where MSWT achieves 76.5% and 80.6% improvements over the second-best method. This difference is reflected in the rollout predictions. As shown in Fig. 2a (and Appendix Fig. 5), MSWT attains the lowest predictive error at step 1 in regions of strong shear, where small-scale structures are most challenging to accurately capture. These errors gradually propagate to larger scales by step 64. In contrast, HFS produces physically plausible rollouts but accumulates large-scale errors, while FNO yields unphysical solutions by step 64. The enstrophy power spectrum in Fig. 2b echoes this observation: MSWT closely follows the ground-truth spectrum whereas the baselines overestimate energy, even at low wavenumbers, indicating distortion at large scales that is induced by a failure to accurately represent the small scales at prior steps in the rollout. Overall, these results highlight the effectiveness of the proposed wavelet-based sampling and wavelet attention in tracking high-frequency (small-scale) patterns over long rollouts.

### 5.2. Shallow Water Equation (SWE)

Here we test the model's ability to stably roll out nonlinear wave propagation and long-range spatiotemporal interac-

*Table 2.* SWE short-/long-term evaluation (mean $\pm$ std) for **Rel $L^2$**, **EMLR**, and **EMAE** ($\downarrow$). Best entries highlighted. "–" indicates unstable long rollouts. The last row reports the relative improvement of MSWT over the second-best stable model.

| | Rel $L^2$ | | | EMLR | | | EMAE | | |
|---|---|---|---|---|---|---|---|---|---|
| Model | step 1 | step 41 | step 81 | step 1 | step 41 | step 81 | step 1 | step 41 | step 81 |
| FNO | $0.09 \pm 0.00$ | $0.33 \pm 0.01$ | $0.58 \pm 0.03$ | $0.11 \pm 0.00$ | $0.20 \pm 0.01$ | $0.30 \pm 0.01$ | $0.11 \pm 0.00$ | $0.20 \pm 0.02$ | $0.39 \pm 0.02$ |
| Unet | $0.05 \pm 0.00$ | – | – | $0.02 \pm 0.00$ | – | – | $0.02 \pm 0.00$ | – | – |
| WNO | $1.00 \pm 0.00$ | $1.00 \pm 0.00$ | $1.00 \pm 0.00$ | – | – | – | $1.00 \pm 0.00$ | $1.00 \pm 0.00$ | $1.00 \pm 0.00$ |
| SAOT | $0.07 \pm 0.00$ | $0.72 \pm 0.24$ | $1.25 \pm 0.51$ | $0.03 \pm 0.00$ | $0.51 \pm 0.18$ | $0.97 \pm 0.43$ | $0.03 \pm 0.00$ | $1.12 \pm 0.55$ | $3.27 \pm 2.76$ |
| HFS | $0.05 \pm 0.00$ | $0.33 \pm 0.03$ | $0.66 \pm 0.08$ | $\mathbf{0.01 \pm 0.00}$ | $0.25 \pm 0.05$ | $0.51 \pm 0.13$ | $\mathbf{0.01 \pm 0.00}$ | $0.36 \pm 0.11$ | $1.09 \pm 0.44$ |
| MSWT | $\mathbf{0.05 \pm 0.00}$ | $\mathbf{0.22 \pm 0.02}$ | $\mathbf{0.42 \pm 0.07}$ | $0.02 \pm 0.00$ | $\mathbf{0.13 \pm 0.00}$ | $\mathbf{0.22 \pm 0.01}$ | $0.02 \pm 0.00$ | $\mathbf{0.14 \pm 0.01}$ | $\mathbf{0.27 \pm 0.03}$ |
| Rel. improv. (%) | 0.0 | 33.3 | 27.6 | 0.0 | 35.0 | 26.7 | 0.0 | 30.0 | 30.8 |

tions. This task is also widely used as a surrogate benchmark for weather forecasting. We compare the Relative $L^2$ norm, EMLR, and EMAE of the evaluated methods at three time steps (1, 41, and 81) in Table 2 (SMLR and SMAE results are reported in Appendix Table 7). We also present predictions of selected methods in Fig. 3 (the corresponding power spectra are shown in Appendix Fig. 16).

We see that MSWT achieves the second-best one-step metrics, slightly behind HFS, but delivers substantial gains in long-horizon predictive rollouts at steps 41 and 81. It improves over the second-best stable baseline (FNO) by roughly 30% in Relative $L^2$, EMLR, and EMAE, consistently demonstrating lower error accumulation and better long-term stability. The predictive error in Fig. 3 further shows that for long rollouts MSWT exhibits smaller-scale errors than FNO and HFS. In addition, MSWT avoids the boundary artifacts observed in HFS, which arise from their non-periodic padding in convolutional operators.

Regarding the power spectrum performance, all methods follow the ground-truth kinetic energy and enstrophy closely at step 1 (Appendix Fig. 12). By step 41, MSWT shows mild oversmoothing, with reduced spectral energy at low wavenumbers (1–6 in Appendix Fig. 14), while baselines such as HFS exhibit pronounced overestimation at high wavenumbers (greater than 20). At step 81, MSWT achieves the smallest bias in the low-frequency spectrum (Appendix Fig. 16), suggesting improved preservation of both low- and high-frequency content under long-horizon prediction with the proposed wavelet-based architecture.

### 5.3. ERA5 Climate Reanalysis

In this real-world benchmark, we evaluate the model's ability to remain stable over decade-scale autoregressive rollouts while reproducing climatological statistics. The model is trained using a spherical weighted $L^2$ loss on the Legendre–Gauss grid, together with a spectral regularization term. Starting from an out-of-sample initial condition, we run the rollout predictions for 10 years by providing the external forcing variables as the input and autoregressively predict the variables for the next timestep. We then compute the average over all the rollout years and compare them against the

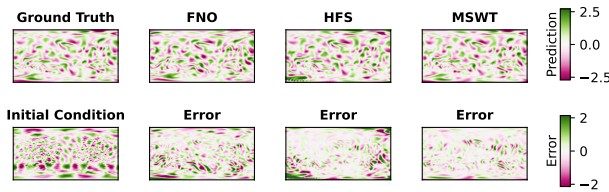

*Figure 3.* Prediction and error comparisons on the SWE dataset at $t = 81$. More detailed comparisons and the power spectrum results can be found in the Appendix B.3.1.

*Table 3.* Comparison between LUCIE and MSWT using RMSE (mean $\pm$ std over 5 random seeds). Variables include units in brackets. The lowest RMSE ($\downarrow$) is highlighted.

| Variable | LUCIE | MSWT |
|---|---|---|
| temperature (K) | $1.14 \pm 0.76$ | $\mathbf{0.86 \pm 0.28}$ |
| humidity (g/kg) | $0.36 \pm 0.18$ | $\mathbf{0.32 \pm 0.11}$ |
| zonal wind (m/s) | $2.45 \pm 1.59$ | $\mathbf{1.92 \pm 0.17}$ |
| meridional wind (m/s) | $1.04 \pm 0.40$ | $\mathbf{0.87 \pm 0.07}$ |
| surface pressure (hPa) | $2.97 \pm 2.56$ | $\mathbf{1.31 \pm 0.12}$ |
| precipitation (mm/d) | $0.73 \pm 0.20$ | $\mathbf{0.66 \pm 0.13}$ |

ground-truth climatology (see Fig. 4 and Appendix Fig. 17–18). We report the RMSE of climatology errors in Table 3 (min/max bias results are provided in Appendix Table 8). We also evaluate zonal means (Appendix Fig. 19) and power spectra (Appendix Fig. 20).

MSWT consistently attains smaller errors than LUCIE across variables, yielding lower RMSE and reduced bias magnitudes, which indicates improved long-term stability under free-running rollouts. We observe notably smaller surface-pressure bias near the polar regions (60°N–90°N and 60°S–90°S in Fig. 4), and reduced meridional-wind bias over the South Pole (around 90°S; Appendix Fig. 19) where LUCIE exhibits extreme deviations. MSWT also has smaller zonal errors for precipitation near the equator, but LUCIE shows smaller global min/max bias (Appendix Table 8). Finally, MSWT yields improved power spectra with reduced spectral distortion (Appendix Fig. 20). Overall, these results highlight the stability and accuracy of MSWT on a real-world climate setting under decade-long rollouts.

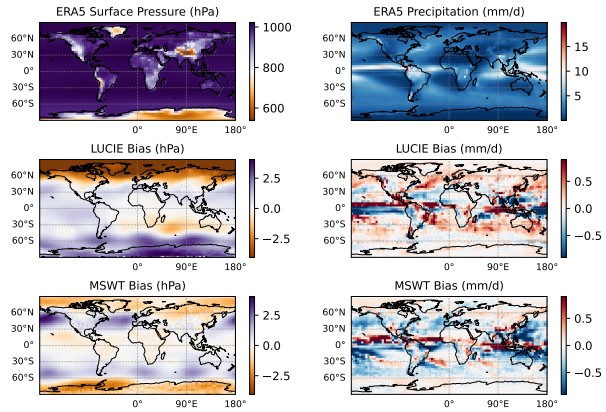

(a) Surface pressure (hPa)     (b) Precipitation (mm/d)

*Figure 4.* Ensemble-mean annual climatology bias of LUCIE and MSWT relative to ERA5 over the time period 2000–2010. The climatology is averaged over 5 ensemble members.

## 5.4. Parameter Sensitivity Analysis

We examine how the patch size used in patch tokenization affects model performance. Specifically, we evaluate patch sizes in $2, 4, 8$ with a fixed wavelet downsampling depth of $L = 3$ on the CKF benchmark. The results in Table 6 in Appendix B.5.1 show that increasing the patch size degrades both short-term and long-term rollout accuracy, with the largest deterioration observed for a patch size of 8. We attribute this behavior to the relatively low input resolution $(64 \times 64)$, for which large patches lead to overly aggressive compression. Embedding such large patches with a simple linear tokenizer can cause substantial information loss, which in turn amplifies error accumulation and induces long-term instability. We also perform an ablation on the number of model parameters comparing FNO and MSWT in Appendix B.5.2, and runtime comparisons across all models are presented in Appendix B.6 with both training and inference times.

## 6. Conclusion

We have proposed the **Multi-Scale Wavelet Transformer (MSWT)**, a new operator-learning framework for surrogate modelling of dynamical systems. After patch tokenization, the model adopts a U-shaped architecture equipped with wavelet attention and multi-scale wavelet downsampling and upsampling, explicitly preserving both low- and high-frequency features across scales while maintaining computational efficiency. Experiments on challenging PDE simulation benchmarks demonstrate the superiority of MSWT in both short- and long-term rollouts, achieving substantial improvements in relative $L^2$ error and spectral metrics (60% for the chaotic Kolmogorov flow benchmark and 30% for the shallow water equation benchmark). Furthermore, real-world validation on the ERA5 climate reanalysis benchmark highlights the long-term stability of MSWT under decade-scale autoregressive rollouts, where it attains consistently lower climatological bias than state-of-the-art methods.

## Impact Statement

Our method improves the efficiency and long-horizon stability of learned surrogate models for spatiotemporal dynamical systems, enabling faster rollouts for tasks such as uncertainty quantification, sensitivity analysis, inverse problems, and rapid what-if exploration in scientific simulation and forecasting pipelines. This can reduce compute cost and inference time, potentially widening access to high-quality simulation tools and supporting applications such as environmental and climate-risk research or infrastructure planning. However, as a data-driven surrogate, outputs should be accompanied by validation (and ideally uncertainty estimates) when used for high-stakes decisions.

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

## A. Wavelet transforms

In the following, we provide a brief overview on wavelet transforms. We will focus on the real-valued case, which is more applicable to our setting, though wavelets are more generally defined in terms of complex-valued functions. More details about wavelet transforms can be found in well known literary references, such as Meyer (1992) and Mallat (2008).

**Continuous wavelet transform (CWT).** Let $\psi \in L^2(\mathbb{R})$, where $L^2(\mathbb{R})$ denotes the space of square-integrable functions over $\mathbb{R}$, be such that:

$$\int_{-\infty}^{+\infty} \psi(t)\, \mathrm{d}t = 0 \tag{12}$$

$$\int_{-\infty}^{+\infty} \psi^2(t)\, \mathrm{d}t = 1. \tag{13}$$

Then $\psi$ is called an admissible wavelet if:

$$C_\psi = \int_0^\infty \frac{|\widehat{\psi}(\omega)|^2}{\omega}\, \mathrm{d}\omega < \infty, \tag{14}$$

where $\widehat{\psi}$ is the Fourier transform of $\psi$. Considering a function $f \in L^2(\mathbb{R})$, the continuous wavelet transform $\mathcal{W}_\psi$ of $f$ associated with $\psi$ is given by:

$$\mathcal{W}_\psi f(a,b) = \frac{1}{\sqrt{a}} \int_{-\infty}^{+\infty} f(t)\psi\left(\frac{t-b}{a}\right) \mathrm{d}t, \qquad a > 0, \, b \in \mathbb{R}, \tag{15}$$

where $a$ and $b$ represent scaling and shift, respectively. We note that, when applied to wavelet transforms, an admissible wavelet $\psi$ is also called a "mother wavelet" function because we can define scaled and shifted child versions of $\psi$ as:

$$t \in \mathbb{R}, \qquad \psi_{a,b}(t) = \frac{1}{\sqrt{a}}\psi\left(\frac{t-b}{a}\right), \qquad a > 0, \, b \in \mathbb{R}. \tag{16}$$

Hence, we can describe the wavelet transform as the $L^2(\mathbb{R})$ inner product $\mathcal{W}_\psi f(a,b) = \langle f, \psi_{a,b} \rangle$, or equivalently as a convolution $\mathcal{W}_\psi f(a,b) = f * \tilde{\psi}_{a,b}$ between $f$ and $\tilde{\psi}_{a,b}(t) = \psi_{a,b}(-t)$. Given the wavelet transform of any $f \in L^2(\mathbb{R})$, we can reconstruct $f$ via an inverse transform $\mathcal{W}_\psi^{-1}$ as:

$$f(t) = (\mathcal{W}_\psi^{-1}\mathcal{W}_\psi f)(t) = \frac{2}{C_\psi} \int_0^{+\infty} \int_{-\infty}^{+\infty} \mathcal{W}_\psi f(a,b) \frac{\psi_{a,b}(t)}{a^2}\, \mathrm{d}a\, \mathrm{d}b, \qquad t \in \mathbb{R}. \tag{17}$$

See Meyer (1992, Thm. 4.4) for a proof.

**Orthonormal wavelets.** If we are dealing with discrete signals, such as the discretization of a time series or an image over a finite grid, we can define the discrete wavelet transform. In addition to the general conditions that an admissible wavelet must satisfy, in the case of the DWT, the mother wavelet $\psi$ needs to satisfy additional constraints that allow us to form an orthonormal basis for the space of functions we are dealing with (Meyer, 1992). Namely, let scaling and shift follow a dyadic pattern, i.e., $a = 2^{-j}$, $b = k2^{-j}$, for $j, k \in \mathbb{Z}$, we then consider wavelets:

$$\psi_{jk}(t) = 2^{j/2}\psi(2^{j/2}t - k), \qquad j, k \in \mathbb{Z}, \quad t \in \mathbb{R}, \tag{18}$$

such that $\langle \psi_{jk}, \psi_{lm} \rangle = 1$ if $j = k = l = m$, and 0 otherwise. When $\psi$ satisfies these conditions, we have that any $f \in L^2(\mathbb{R})$ can be decomposed as (Meyer, 1992, Ch. 3):

$$f(t) = \sum_{j,k\in\mathbb{Z}} c_{jk}\psi_{jk}(t), \qquad t \in \mathbb{R}, \tag{19}$$

where $c_{jk} = \mathcal{W}_\psi f(2^{-j}, k2^{-j}) = \langle f, \psi_{jk} \rangle$. Hence, $\{\psi_{jk}\}_{j,k\in\mathbb{Z}}$ form an orthonormal basis of $L^2(\mathbb{R})$. Moreover, each $\psi_{jk}$ can be shown to concentrate on the interval $[k2^{-j}, (k+1)2^{-j})$, rapidly vanishing outside of it. We can then interpret the formulation above as decomposing a function $f$ over several series $\mathcal{I}_j := \{[k2^{-j}, (k+1)2^{-j})\}_{k\in\mathbb{Z}}$ of adjacent (disjoint) intervals spanning $\mathbb{R}$, each series taking steps at an arbitrarily fine scale $2^{-j}$. The value of $f$ within a given interval $[k2^{-j}, (k+1)2^{-j})$ at a given scale is determined by the wavelet transform $\mathcal{W}_\psi f(2^{-j}, k2^{-j})$. Thus, we have a multiresolution decomposition of the original signal $f$.

**Scaling function.** A scaling function $\phi$ represents an aggregation of wavelets at scales larger than 1 and is used when we only know $\mathcal{W}_\psi$ up to a truncated scale (Mallat, 2008, p. 76). We can define $\phi$ via its Fourier transform $\widehat{\psi}$, which must satisfy:

$$|\widehat{\phi}(\omega)|^2 = \int_1^{+\infty} \frac{|\widehat{\psi}(s\omega)|^2}{s} \, \mathrm{d}s = \int_\omega^{+\infty} \frac{|\widehat{\psi}(\xi)|^2}{\xi} \, \mathrm{d}\xi, \tag{20}$$

and the complex phase for $\widehat{\phi}$ can be arbitrarily chosen. It follows that $\|\phi\| = 1$ and due to the admissibility condition on $\psi$,

$$\lim_{\omega \to 0} |\widehat{\phi}(\omega)|^2 = C_\psi. \tag{21}$$

Thus, convolving with $\phi$ is equivalent to passing a signal through a low-pass filter, i.e., an averaging/smoothing effect. In addition, we define the dyadic scaled and shifted versions of $\phi$ as:

$$\phi_{jk}(t) = 2^{j/2}\phi(2^j t - k), \qquad j, k \in \mathbb{Z}. \tag{22}$$

**Discrete wavelet transform (DWT).** Consider a continuous signal $f : [0,1] \to \mathbb{R}$, and let $\boldsymbol{f} := [f(t_i)]_{i=0}^{N-1}$ denote its discretization over a set of $N$ regularly spaced points $\{t_i\}_{i=0}^{N-1} \subset \mathbb{R}$, where $t_i = \frac{i}{N}$, assuming $N = 2^J$, for some $J \in \mathbb{N}$. Given a mother wavelet $\psi$ satisfying the orthogonality constraints above, the DWT decomposes the signal $\boldsymbol{f}$ as:

$$\mathcal{W}_\psi \boldsymbol{f} = (\alpha_{J,0}, \{d_{jk}\}_{j,k}), \tag{23}$$

where:

$$d_{jk} = \boldsymbol{f} \cdot \boldsymbol{\psi}_{jk}, \qquad j \in \{1, \ldots, J\}, \quad k \in \{0, \ldots, 2^{J-j} - 1\} \tag{24}$$
$$\alpha_{J,0} = \boldsymbol{f} \cdot \boldsymbol{\phi}_{J,0}, \tag{25}$$

with $\boldsymbol{\psi}_{jk} = [\psi_{jk}(t_i)]_{i=0}^{N-1}$ and $\boldsymbol{\phi}_{J,0} = [\phi_{J,0}(t_i)]_{i=0}^{N-1}$ (cf. Equation 22).

**Fast DWT with Haar wavelets.** The Haar wavelet is given by:

$$\psi(t) = \mathbb{1}_{[0,1/2)}(t) - \mathbb{1}_{[1/2,1)}(t) \tag{26}$$
$$\phi(t) = \mathbb{1}_{[0,1)}(t), \tag{27}$$

for $t \in \mathbb{R}$, where $\mathbb{1}_{\mathcal{I}}(t) = 1$ if $t \in \mathcal{I}$, and 0 otherwise. Their definition greatly simplifies the computation of the DWT via a recursive process using the matrix:

$$\boldsymbol{H} = \frac{1}{\sqrt{2}} \begin{bmatrix} 1 & 1 \\ 1 & -1 \end{bmatrix}, \tag{28}$$

where the upper row represents a low-pass filter (i.e., smoothing) $\boldsymbol{g} = \frac{1}{\sqrt{2}}[1, 1]$, producing local averages, and the lower row is a high-pass filter (i.e., difference between consecutive points) $\boldsymbol{h} = \frac{1}{\sqrt{2}}[1, -1]$, producing "details" of the signal. One can then operate in pairs of points, starting with $\boldsymbol{\alpha}_0 = \boldsymbol{f}$, and iterating:

$$\begin{bmatrix} \alpha_{jk} \\ d_{jk} \end{bmatrix} = \boldsymbol{H} \begin{bmatrix} \alpha_{j-1,2k} \\ \alpha_{j-1,2k+1} \end{bmatrix}, \qquad j \in \{1, \ldots, J\}, \quad k \in \{0, \ldots, N/2 - 1\}. \tag{29}$$

In this recursion, we keep the details $d_{jk}$ and process the resulting averages $\alpha_{jk}$. At the end of the process, $\alpha_{J,0}$ represents a global average and $d_{jk}$ correspond to details of the signal under windows of increasing length $2^j$, for $j \in \{1, \ldots, J\}$, recalling our assumption that $N = 2^J$.

Note that in this 1-dimensional case, we are basically splitting the signal into two components $(\boldsymbol{\alpha}_j, \boldsymbol{d}_j)$ at every step of the transform's recursion. The extension to 2D follows a similar process, however, splitting the signal into four components, considering correlations across the signal's components, as described in the main paper.

## B. Experiments

### B.1. Criteria

**Rel $L^2$** (Li et al., 2021): Relative $L^2$ norm is used to compare the prediction $\hat{\boldsymbol{u}}$ with the ground truth solutions $\boldsymbol{u}$:

$$\mathrm{Rel}L^2 = \frac{\|\hat{\boldsymbol{u}} - \boldsymbol{u}\|_2}{\|\boldsymbol{u}\|_2} \tag{30}$$

**SMAE:**   Spectral Mean Absolute Error compares the predicted kinetic energy spectrum $\hat{E}_k$ with the ground-truth spectrum $E_k$ by the mean absolute error over Fourier modes. At a given time step, we first obtain velocity fields from vorticity, $(\hat{u}_x, \hat{u}_y) = \mathcal{V}(\hat{\omega})$, $(u_x, u_y) = \mathcal{V}(\omega)$, and compute the (2D) spectral energy per mode $k$ via

$$\boldsymbol{E}_k = \frac{1}{2}\frac{|\mathcal{F}(u_x)_k|^2 + |\mathcal{F}(u_y)_k|^2}{N^2}, \qquad \hat{\boldsymbol{E}}_k = \frac{1}{2}\frac{|\mathcal{F}(\hat{u}_x)_k|^2 + |\mathcal{F}(\hat{u}_y)_k|^2}{N^2},$$

where $\mathcal{F}(\cdot)$ denotes the discrete Fourier transform and $N = N_x N_y$. Then

$$\text{SMAE} = \frac{1}{|\mathcal{K}|} \sum_{k \in \mathcal{K}} \left| \frac{\hat{\boldsymbol{E}}_k - \boldsymbol{E}_k}{\boldsymbol{E}_k} \right| \tag{31}$$

Here $\mathcal{K}$ is the set of retained Fourier modes.

**SMLR:**   Spetral Mean Log Ratio compares $\hat{\boldsymbol{E}}_k$ and $\boldsymbol{E}_k$ via the mean absolute log-ratio over Fourier modes:

$$\text{SMLR} = \frac{1}{|\mathcal{K}|} \sum_{k \in \mathcal{K}} \left| \log\left( \frac{\hat{\boldsymbol{E}}_k}{\boldsymbol{E}_k} \right) \right| \tag{32}$$

**EMAE:**   Analogously, we define Enstrophy Mean Absolute Error on the enstrophy spectrum. Given vorticity fields $\omega$ and $\hat{\omega}$, we compute the (2D) enstrophy spectrum per mode $k$ as

$$\boldsymbol{Z}_k = \frac{|\mathcal{F}(\omega)_k|^2}{N^2}, \qquad \hat{\boldsymbol{Z}}_k = \frac{|\mathcal{F}(\hat{\omega})_k|^2}{N^2},$$

and define

$$\text{EMAE} = \frac{1}{|\mathcal{K}|} \sum_{k \in \mathcal{K}} \left| \frac{\hat{\boldsymbol{Z}}_k - \boldsymbol{Z}_k}{\boldsymbol{Z}_k} \right| \tag{33}$$

**EMLR:**   The corresponding mean absolute log-ratio error on the enstrophy spectrum is

$$\text{EMLR} = \frac{1}{|\mathcal{K}|} \sum_{k \in \mathcal{K}} \left| \log\left( \frac{\hat{\boldsymbol{Z}}_k}{\boldsymbol{Z}_k} \right) \right| \tag{34}$$

### B.2. Chaotic Kolmogorov Flow

The Chaotic Kolmogorov Flow (Li et al., 2024) is described by the Navier-Stokes equation in two dimensional incompressible flows. We utilize the vorticity formulation to evaluate the baselines:

$$\frac{\partial \omega(\boldsymbol{x}, t)}{\partial t} + \boldsymbol{u}(\boldsymbol{x}, t) \cdot \nabla \omega(\boldsymbol{x}, t) = \frac{1}{Re} \nabla^2 \omega(\boldsymbol{x}, t) + \boldsymbol{f} \quad \boldsymbol{x} \in [0, 2\pi]^2, \quad t \in [0, 0.5] \tag{35}$$

where $\boldsymbol{u}(\boldsymbol{x}, t)$ is the velocity field, $w(x, t) = \frac{\partial u_2}{\partial x_1} + \frac{\partial u_1}{\partial x_2}$ is the scalar vorticity field. We choose the Reynolds number $Re = 500$ and set the external forcing $\boldsymbol{f} = (0, -4\cos(4x_2))$. Equation 35 was solved on a spatial grid of $64 \times 64$ points with the periodic boundary condition and a temporal horizon of 65 time steps. We randomized 4000 initial conditions for training and left out 100 new samples for testing.

#### B.2.1. BASELINE MODEL HYPERPARAMETER SETUP

We train all models using Adam with $(\beta_1, \beta_2) = (0.9, 0.999)$ and initial learning rate $10^{-3}$, using batch size 100 for 100,000 training epochs/iterations. We apply a MultiStepLR schedule and decay factor $\gamma = 0.5$. Optimization uses the relative $L^2$ norm.

#### B.2.2. RESULTS

We present the predictions and the corresponding errors of the five models: FNO, Unet, SAOT, HFS and MSWT. We also illustrate the kinetic energy and enstrophy spectra at multiple steps 1, 30, and 64.

*Table 4.* Hyperparameter settings and parameter counts for baselines and MSWT.

| Model | Setting | # params (M) |
|---|---|---|
| FNO | $N_{\text{layers}}=5$, $n_{\text{hidden}}=64$, `truncation_mode=16` | 16.8 |
| Unet | $N_{\text{layers}}=4$, $n_{\text{hidden}}=[16, 32, 64, 256]$ | 12.5 |
| WNO | $N_{\text{layers}}=4$, $n_{\text{hidden}}=96$, `multiscale_levels=3` | 14.5 |
| SAOT | $N_{\text{layers}}=5$, $n_{\text{hidden}}=384$ | 14.0 |
| HFS | $N_{\text{layers}}=5$, $n_{\text{hidden}}=[32, 64, 128, 256, 256]$ | 13.1 |
| MSWT | $N_{\text{layers}}=4$, $n_{\text{hidden}}=[64, 64, 128, 512]$ | 13.5 |

*Table 5.* CKF long-term evaluation (mean $\pm$ std) for **SMLR** and **SMAE**. Best MSWT entries highlighted.

| Model | SMLR step 1 | step 30 | step 64 | SMAE step 1 | step 30 | step 64 |
|---|---|---|---|---|---|---|
| FNO | $0.08 \pm 0.00$ | $0.53 \pm 0.01$ | $0.64 \pm 0.02$ | $0.07 \pm 0.00$ | $0.83 \pm 0.02$ | $1.08 \pm 0.06$ |
| Unet | $0.06 \pm 0.06$ | $1.42 \pm 0.27$ | $6.09 \pm 1.72$ | $0.87 \pm 1.87$ | – | – |
| WNO | $0.37 \pm 0.00$ | – | – | $5.22 \pm 0.00$ | – | – |
| SAOT | $0.04 \pm 0.01$ | $0.54 \pm 0.25$ | $0.93 \pm 0.50$ | $0.04 \pm 0.01$ | $0.88 \pm 0.61$ | $2.00 \pm 1.79$ |
| HFS | $0.04 \pm 0.00$ | $0.58 \pm 0.05$ | $0.86 \pm 0.06$ | $0.04 \pm 0.00$ | $1.02 \pm 0.13$ | $1.96 \pm 0.34$ |
| MSWT | $\mathbf{0.03 \pm 0.00}$ | $\mathbf{0.16 \pm 0.02}$ | $\mathbf{0.20 \pm 0.02}$ | $\mathbf{0.02 \pm 0.00}$ | $\mathbf{0.15 \pm 0.01}$ | $\mathbf{0.21 \pm 0.02}$ |

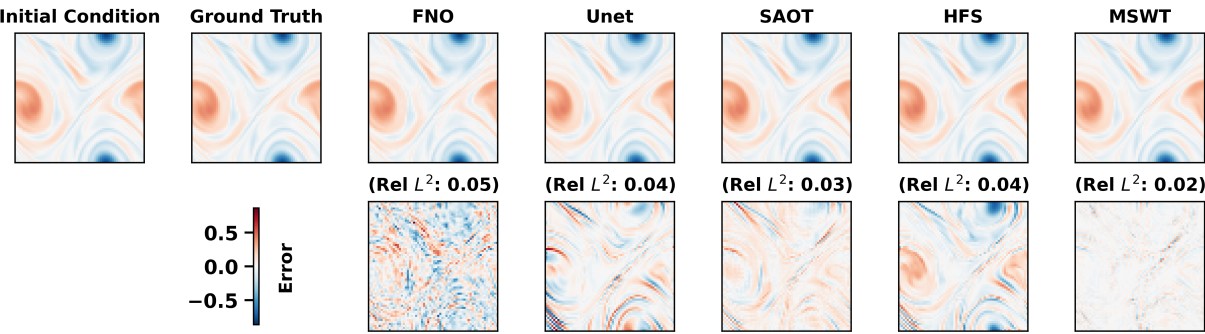

*Figure 5.* Chaotic Kolmogorov Flow, prediction and error comparison, rollout step =1

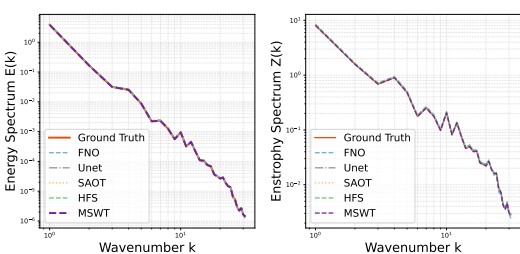

*Figure 6.* Chaotic Kolmogorov Flow, kinetic energy spectrum and enstrophy spectrum, rollout step =1

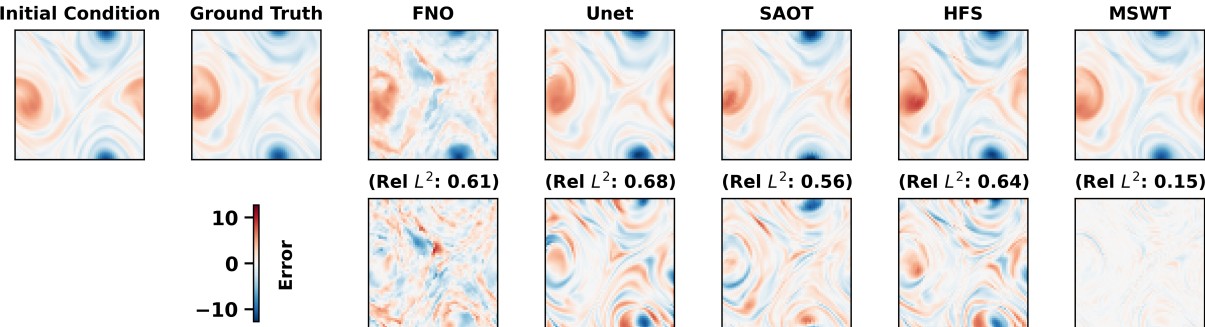

*Figure 7.* Chaotic Kolmogorov Flow, prediction and error comparison, rollout step =30

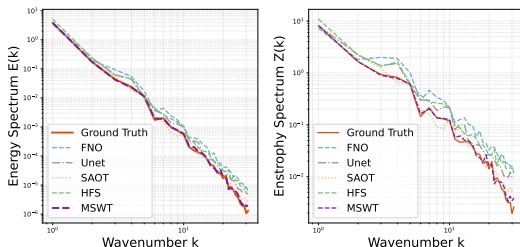

*Figure 8.* Chaotic Kolmogorov Flow, kinetic energy spectrum and enstrophy spectrum, rollout step =30

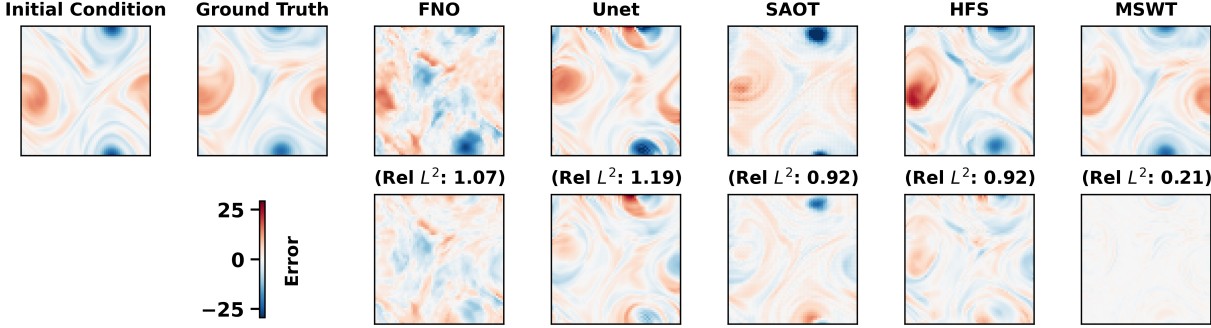

*Figure 9.* Chaotic Kolmogorov Flow, prediction and error comparison, rollout step =64

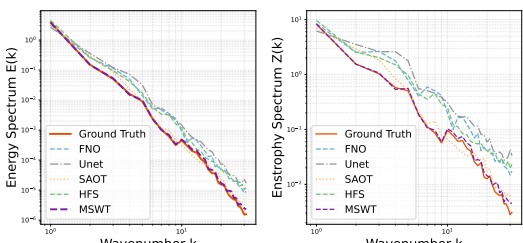

*Figure 10.* Chaotic Kolmogorov Flow, kinetic energy spectrum and enstrophy spectrum, rollout step =64

*Table 6.* Effect of patch size on rollout metrics.

| Patch size | Rel $L^2$ | | | SMLR | | | EMLR | | | SMAE | | | EMAE | | |
|---|---|---|---|---|---|---|---|---|---|---|---|---|---|---|---|
| | step 1 | step 30 | step 64 | step 1 | step 30 | step 64 | step 1 | step 30 | step 64 | step 1 | step 30 | step 64 | step 1 | step 30 | step 64 |
| 2 | 0.0258 | 0.3286 | 0.5776 | 0.0362 | 0.2180 | 0.3116 | 0.0361 | 0.2176 | 0.3111 | 0.0345 | 0.2136 | 9.9425 | 0.0344 | 0.2132 | 9.7618 |
| 4 | 0.0298 | 0.3429 | 0.5535 | 0.0223 | 0.4418 | 0.6833 | 0.0223 | 0.4421 | 0.6832 | 0.0219 | 0.8269 | 1.4697 | 0.0219 | 0.8282 | 1.4705 |
| 8 | 0.0780 | 0.6434 | 0.9466 | 0.1516 | 1.0873 | 1.5822 | 0.1517 | 1.0864 | 1.5813 | 0.1250 | 2.6674 | 5.3487 | 0.1250 | 2.6637 | 5.3441 |

*Table 7.* SWE long-term evaluation (mean $\pm$ std) for **SMLR** and **SMAE**. Best MSWT entries highlighted.

| Model | SMLR step 1 | SMLR step 41 | SMLR step 81 | SMAE step 1 | SMAE step 41 | SMAE step 81 |
|---|---|---|---|---|---|---|
| FNO | $0.11 \pm 0.00$ | $0.20 \pm 0.01$ | $0.30 \pm 0.01$ | $0.11 \pm 0.00$ | $0.20 \pm 0.02$ | $0.39 \pm 0.02$ |
| Unet | $0.02 \pm 0.00$ | $-$ | $-$ | $0.02 \pm 0.00$ | $-$ | $-$ |
| WNO | $-$ | $-$ | $-$ | $1.00 \pm 0.00$ | $1.00 \pm 0.00$ | $1.00 \pm 0.00$ |
| SAOT | $0.03 \pm 0.00$ | $0.51 \pm 0.18$ | $0.97 \pm 0.43$ | $0.03 \pm 0.00$ | $1.14 \pm 0.57$ | $3.32 \pm 2.81$ |
| HFS | $\mathbf{0.01 \pm 0.00}$ | $0.25 \pm 0.05$ | $0.52 \pm 0.13$ | $\mathbf{0.01 \pm 0.00}$ | $0.37 \pm 0.11$ | $1.10 \pm 0.44$ |
| MSWT | $0.02 \pm 0.00$ | $\mathbf{0.13 \pm 0.00}$ | $\mathbf{0.22 \pm 0.02}$ | $0.02 \pm 0.00$ | $\mathbf{0.15 \pm 0.01}$ | $\mathbf{0.28 \pm 0.03}$ |

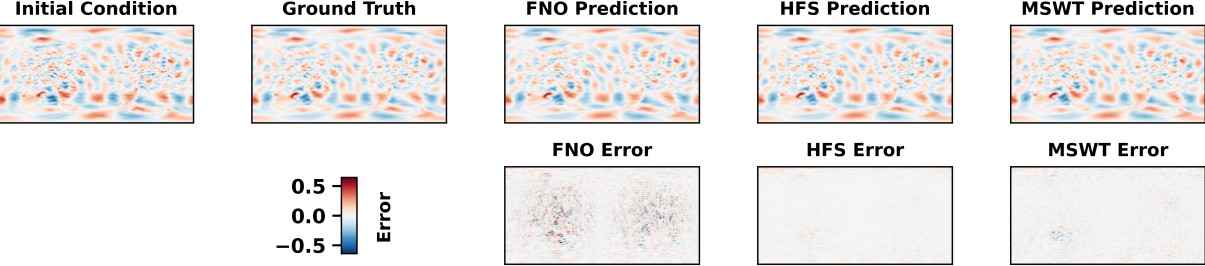

*Figure 11.* Shallow water equation, prediction and error comparison, rollout step =1

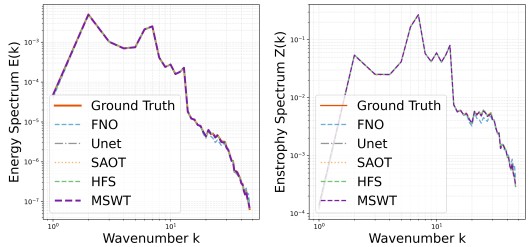

*Figure 12.* Shallow water equation, kinetic energy spectrum and enstrophy spectrum, rollout step =1

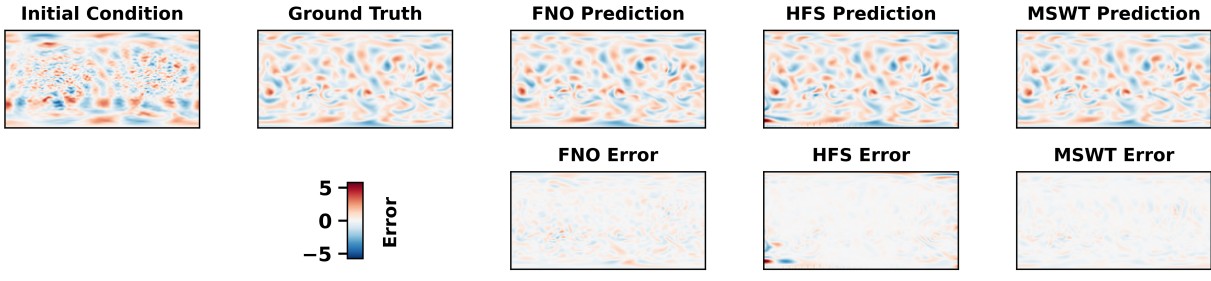

*Figure 13.* Shallow water equation, prediction and error comparison, rollout step =41

## B.3. Shallow Water Equation

We use the PDEArena ShallowWater-2D dataset (Gupta & Brandstetter, 2022) (https://huggingface.co/datasets/pdearena/ShallowWater-2D ), defined on a $96 \times 192$ spatial grid. For our setup, we use pressure and vorticity as input variables and evaluate long-horizon stability over trajectories of 87 autoregressive rollout steps.

### B.3.1. RESULTS

We present the predictions and the corresponding errors of the three models: FNO, HFS and MSWT. We also illustrate the kinetic energy and enstrophy spectra at multiple steps 1, 41, and 81.

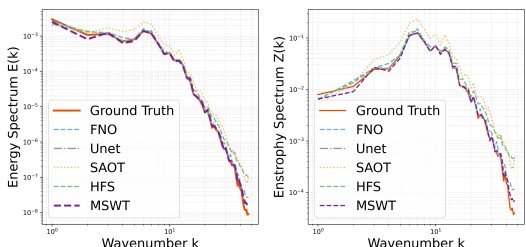

*Figure 14.* Shallow water equation, kinetic energy spectrum and enstrophy spectrum, rollout step =41

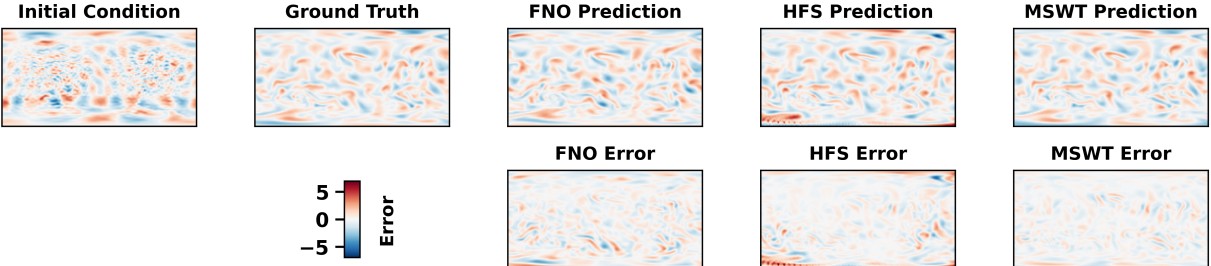

*Figure 15.* Shallow water equation, prediction and error comparison, rollout step =81

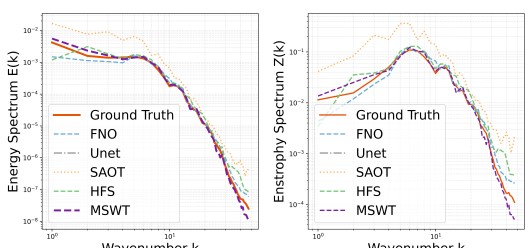

*Figure 16.* Shallow water equation, kinetic energy spectrum and enstrophy spectrum, rollout step =81

## B.4. ERA5

For climate-style evaluation, we follow the LUCIE setup based on coarse-resolution ERA5 fields regridded to a T30 Gaussian grid adapted from their dataset (Guan et al., 2025). The data are sampled every 6 hours from January 2000 to April 2011. The prognostic state includes five variables: temperature, specific humidity, surface pressure and zonal/meridional wind; following LUCIE, we additionally use total incoming solar radiation (TISR) and orography as external inputs to improve long-term stability and capture the seasonal cycle. The diagnostic variable for the predictive output is precipitation. We use the first 16,000 samples for training and the following 100 samples for validation, out of a total of 16,538 data points. The learning task during model training is next-step prediction, i.e., predicting the state at time $t + \Delta t$ from the state at time $t$ with $\Delta t = 6$ hours. For testing, we initialize from an out-of-sample initial condition (i.e., after 2010) and perform a free-running rollout for the next 10 years (14,600 autoregressive steps). We then evaluate the long-horizon behavior using climatology-based metrics comparing the predicted 10-year rollout with temporal averages from the 2000-2010 period, noting that the 10-year averages should remain consistent, given that the forcing terms (TISR and orography) are repeated.

### B.4.1. EVALUATION METRICS

**Ensemble mean.** Let $v \in \{\text{temperature, specific humidity, zonal wind}, \ldots, \text{precipitation}\}$ denote all the prognostic and diagnostic variables. Given 6-hourly spatial fields over the 10-year period 2000–2010, we define the climatology (time-mean) of the model rollout and ERA5 at spatial location $\mathbf{s} \in \Omega$ as

$$\bar{x}_{\text{model}}^{v}(\mathbf{s}) \;=\; \frac{1}{T} \sum_{t=1}^{T} x_{\text{model}}^{v}(t, \mathbf{s}), \qquad \bar{x}_{\text{ERA5}}^{v}(\mathbf{s}) \;=\; \frac{1}{T} \sum_{t=1}^{T} x_{\text{ERA5}}^{v}(t, \mathbf{s}), \tag{36}$$

where $T$ is the number of 6-hourly time steps in the evaluation window. For an ensemble of $K$ free-running rollouts, we compute each member's climatology $\bar{x}_{\text{model}}^{v,(k)}(\mathbf{s})$ and form the ensemble-mean climatology:

$$\bar{x}_{\text{ens}}^{v}(\mathbf{s}) \;=\; \frac{1}{K} \sum_{k=1}^{K} \bar{x}_{\text{model}}^{v,(k)}(\mathbf{s}). \tag{37}$$

**Bias and RMSE for different variables.** The climatology bias field is

$$b^{v}(\mathbf{s}) \;=\; \bar{x}_{\text{model}}^{v}(\mathbf{s}) - \bar{x}_{\text{ERA5}}^{v}(\mathbf{s}). \tag{38}$$

We report global minimum, maximum, and mean biases,

$$\text{MinBias}^{v} = \min_{\mathbf{s} \in \Omega} b^{v}(\mathbf{s}), \quad \text{MaxBias}^{v} = \max_{\mathbf{s} \in \Omega} b^{v}(\mathbf{s}), \quad \text{MeanBias}^{v} = \frac{1}{|\Omega|} \sum_{\mathbf{s} \in \Omega} b^{v}(\mathbf{s}), \tag{39}$$

and the RMSE of the climatology bias,

$$\text{RMSE}^{v} = \sqrt{\frac{1}{|\Omega|} \sum_{\mathbf{s} \in \Omega} \left(b^{v}(\mathbf{s})\right)^{2}}. \tag{40}$$

All bias and RMSE metrics are then computed by substituting $\bar{x}_{\text{ens}}^{v}$ for $\bar{x}_{\text{model}}^{v}$ in the definitions above.

**Zonal mean.** Let $\phi$ and $\lambda$ denote latitude and longitude. The zonal-mean climatology is defined as the longitudinal average of the climatological field:

$$\bar{x}_{\text{zon}}^{v}(\phi) \;=\; \frac{1}{2\pi} \int_{0}^{2\pi} \bar{x}^{v}(\phi, \lambda) \, d\lambda, \tag{41}$$

implemented discretely as an average over longitude grid points. We compare $\bar{x}_{\text{zon}}^{v}(\phi)$ for ERA5 and for the model ensemble-mean climatology over 2000–2010.

*Table 8.* Comparison between LUCIE and MSWT (mean ± std over 5 random seeds). Variables include units in brackets. The lowest Min/Max bias and RMSE (↓↓↓) are highlighted.

| Variable | Min bias | | Max bias | | RMSE | |
|---|---|---|---|---|---|---|
| | LUCIE | MSWT | LUCIE | MSWT | LUCIE | MSWT |
| temperature (K) | $-13.06 \pm 19.93$ | $\mathbf{-2.20 \pm 0.53}$ | $5.32 \pm 3.20$ | $\mathbf{2.69 \pm 0.76}$ | $1.14 \pm 0.76$ | $\mathbf{0.86 \pm 0.28}$ |
| humidity (g/kg) | $-2.11 \pm 1.49$ | $\mathbf{-1.02 \pm 0.36}$ | $3.24 \pm 3.42$ | $\mathbf{1.49 \pm 0.33}$ | $0.36 \pm 0.18$ | $\mathbf{0.32 \pm 0.11}$ |
| zonal wind (m/s) | $-8.50 \pm 3.87$ | $\mathbf{-6.42 \pm 1.48}$ | $6.74 \pm 4.76$ | $\mathbf{6.70 \pm 1.34}$ | $2.45 \pm 1.59$ | $\mathbf{1.92 \pm 0.17}$ |
| meridional wind (m/s) | $-4.74 \pm 2.59$ | $\mathbf{-3.39 \pm 1.04}$ | $4.85 \pm 1.64$ | $\mathbf{2.82 \pm 0.36}$ | $1.04 \pm 0.40$ | $\mathbf{0.87 \pm 0.07}$ |
| surface pressure (hPa) | $-11.47 \pm 3.55$ | $\mathbf{-3.93 \pm 0.76}$ | $14.57 \pm 16.49$ | $\mathbf{4.32 \pm 1.18}$ | $2.97 \pm 2.56$ | $\mathbf{1.31 \pm 0.12}$ |
| precipitation (mm/d) | $\mathbf{-4.07 \pm 1.69}$ | $-4.77 \pm 1.18$ | $\mathbf{8.35 \pm 1.03}$ | $17.02 \pm 15.59$ | $0.73 \pm 0.20$ | $\mathbf{0.66 \pm 0.13}$ |

**Power spectrum.** For each variable $v$, we compute the spatial power spectrum of the ensemble-mean climatological field $\bar{x}_{\text{ens}}^v$ in the spectral domain. Let $\hat{x}_\ell^v$ denote the spectral coefficients and let $k$ index the radial wavenumber. The power spectrum is obtained by binning and summing squared coefficients at wavenumber $k$:

$$P^v(k) \;=\; \sum_{\ell:\,\text{wavenumber}(\ell)=k} \left|\hat{x}_\ell^v\right|^2. \tag{42}$$

We compare $P^v(k)$ between MSWT/LUCIE and ERA5 over 2000–2010 to assess spectral fidelity (e.g., oversmoothing or excess small-scale energy).

## B.5. Ablation studies

### B.5.1. ARCHITECTURE COMPONENT ABLATIONS

To investigate the effectiveness of the three main modules separately in our framework: (1) patch tokenizer; (2) wavelet attention block; (3) wavelet-preserving down-/up-sampling, we propose three variants of the model for ablation studies:

- MSWT-V1 (no tokenizer) uses patch size of 1 for the patch tokenizer to learn the point-wise mapping of each point on the grid;

- MSWT-V2 (no attention) removes the wavelet attention block and only keeps the Unet-based architecture with wavelet-preserving down-/up-sampling modules;

- MSWT-V3 (Conv-downsampling) replaces the wavelet-preserving down-/up-sampling modules with standard convolutions or deconvolutions with the stride of 2 and keeps the wavelet attention blocks.

We validate these three variants on the CKF benchmark and show the results in Supplementary Table 9. Removing the patch tokenizer (V1) improves the performance at the expense of efficiency, whereas removing the attention modules (V2) significantly damaged the performance, manifesting the effectiveness of the wavelet attention block. Using the standard strided-convolution or deconvolution (V3) also achieved sub-optimal results, showing the effectiveness of our proposed wavelet-preserving downsampling and upsampling modules.

*Table 9.* Ablation evaluation (lower is better; ↓) for **Rel $L^2$**, **EMLR**, and **EMAE**. Best entries are highlighted.

| Model | Train time | Rel $L^2$ | | | EMLR | | | EMAE | | |
|---|---|---|---|---|---|---|---|---|---|---|
| | | step 1 | step 30 | step 64 | step 1 | step 30 | step 64 | step 1 | step 30 | step 64 |
| MSWT | 1h17′ | 0.02 | 0.21 | 0.35 | 0.02 | 0.15 | 0.20 | 0.02 | 0.15 | 0.21 |
| MSWT-V1 (no tokenizer) | 3h4′ | **0.01** | **0.10** | **0.18** | **0.01** | **0.07** | **0.10** | **0.01** | **0.07** | **0.10** |
| MSWT-V2 (no attention) | 35′ | 0.06 | 0.81 | 1.09 | 0.04 | 0.69 | 1.32 | 0.04 | 1.24 | 3.90 |
| MSWT-V3 (Conv-downsampling) | 1h4′ | 0.03 | 0.51 | 0.71 | 0.03 | 0.32 | 0.39 | 0.03 | 0.39 | 0.51 |

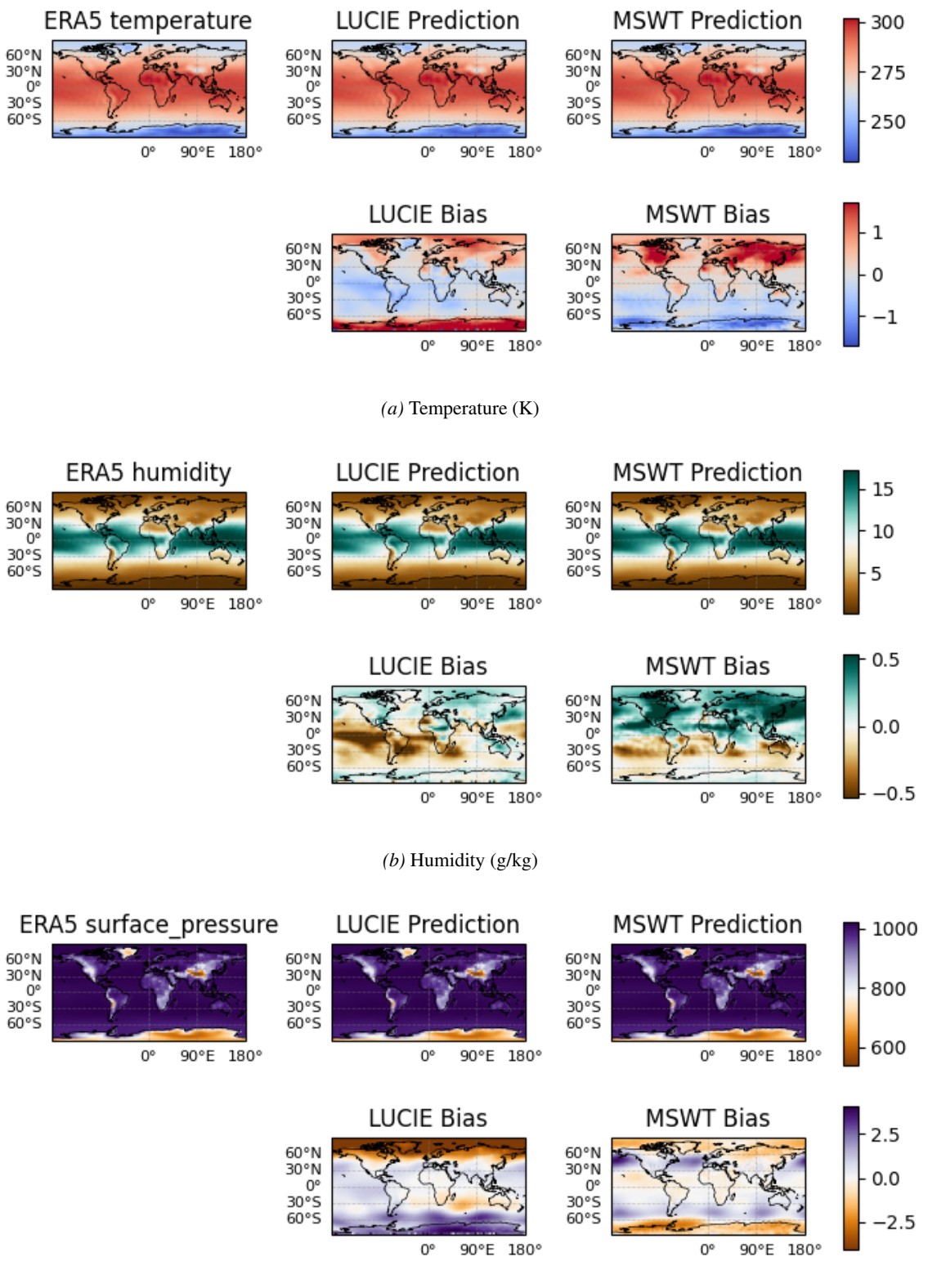

*(a)* Temperature (K)

*(b)* Humidity (g/kg)

*(c)* Surface pressure (hPa)

*Figure 17.* Ensemble-mean annual climatology bias of LUCIE and MSWT (relative to ERA5, 2000–2010). The climatology is averaged over 10 years of simulation and 5 ensemble members.

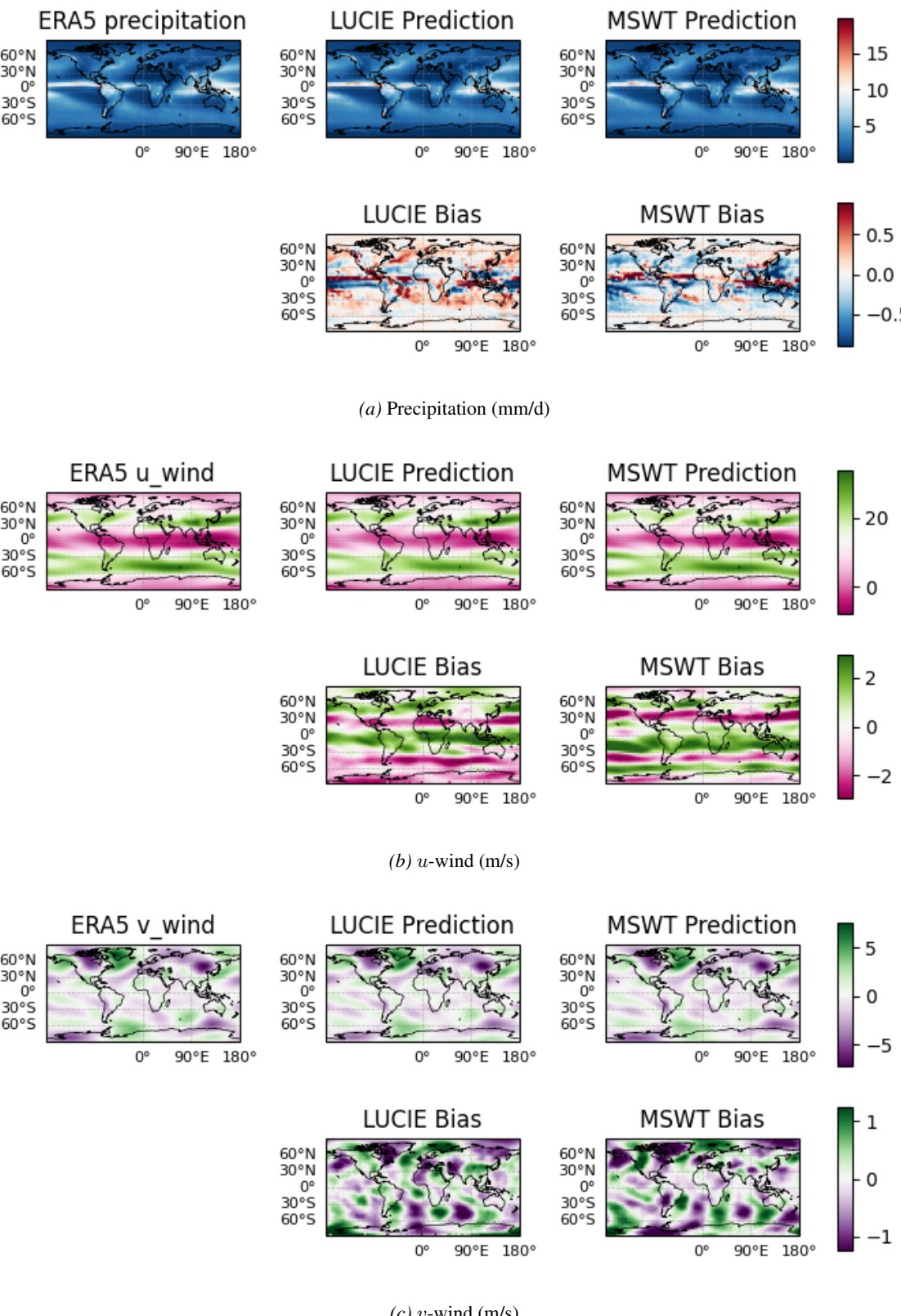

*Figure 18.* Ensemble-mean annual climatology bias of LUCIE and MSWT (relative to ERA5, 2000–2010). The climatology is averaged over 10 years of simulation and 5 ensemble members.

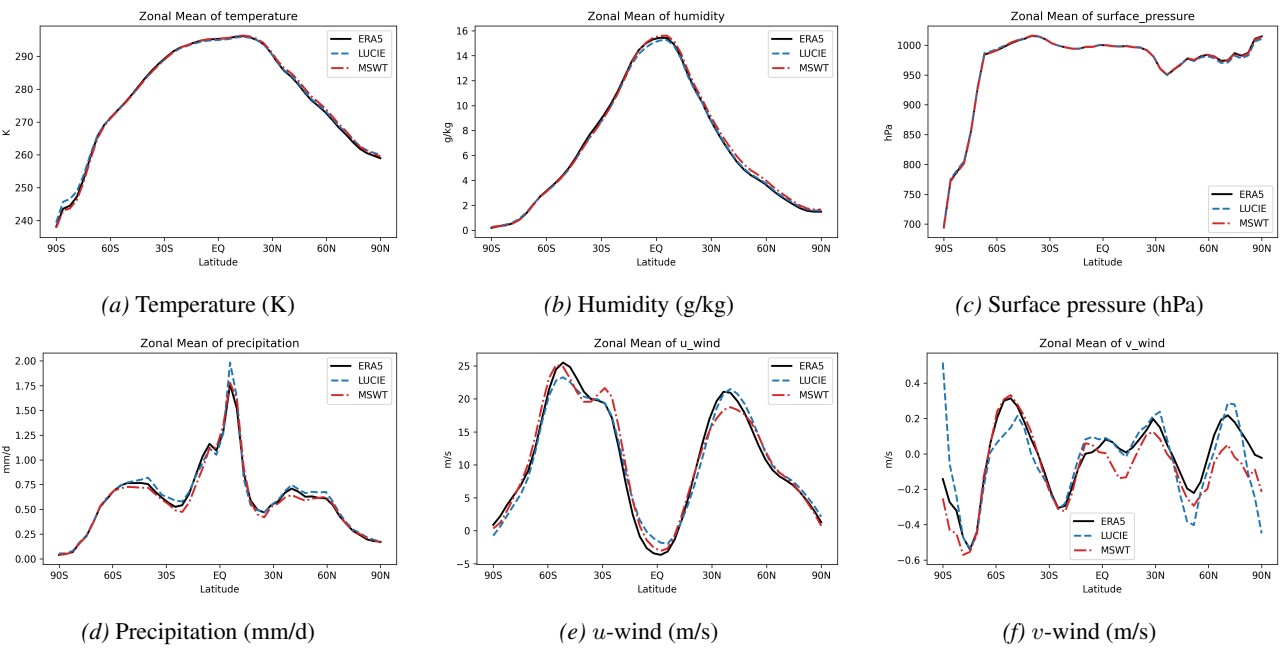

*(a)* Temperature (K)  *(b)* Humidity (g/kg)  *(c)* Surface pressure (hPa)

*(d)* Precipitation (mm/d)  *(e)* $u$-wind (m/s)  *(f)* $v$-wind (m/s)

*Figure 19.* Zonal-mean climatology of the ensemble-mean rollouts (LUCIE, MSWT) compared with ERA5 over 2000–2010.

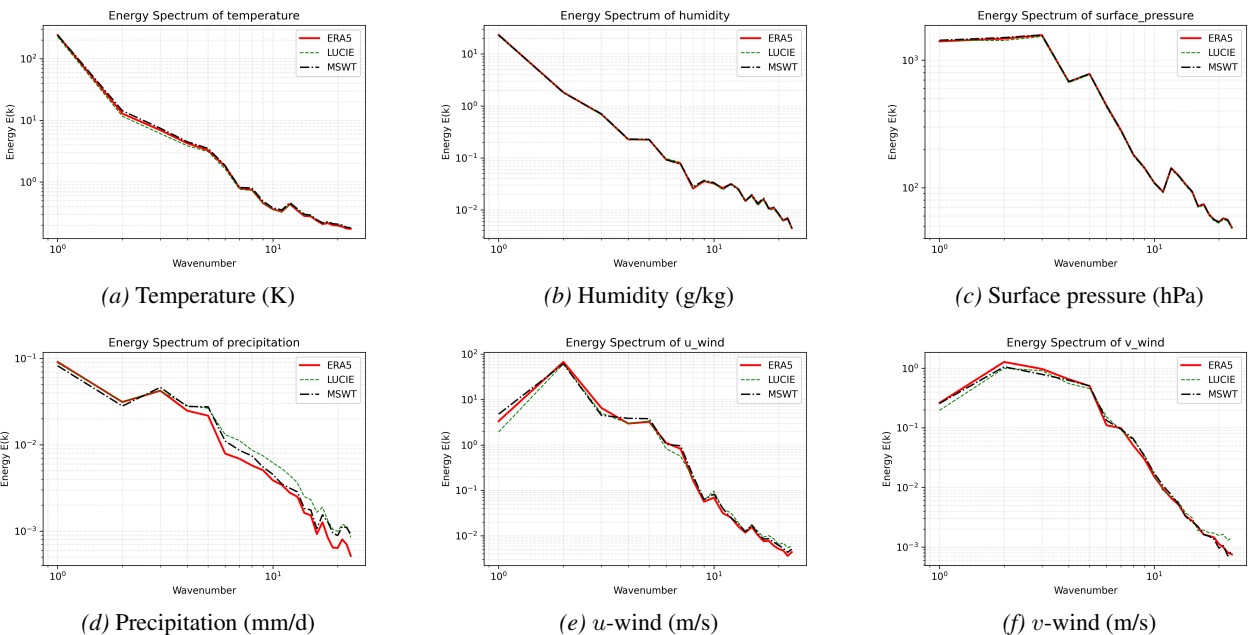

*(a)* Temperature (K)  *(b)* Humidity (g/kg)  *(c)* Surface pressure (hPa)

*(d)* Precipitation (mm/d)  *(e)* $u$-wind (m/s)  *(f)* $v$-wind (m/s)

*Figure 20.* Power (energy) spectra of ensemble-mean rollouts (LUCIE, MSWT) compared with ERA5 over 2000–2010.

*Table 10.* CKF parameter-efficiency ablation. Rollout relative $L_2$ error is measured at the final predictive step in the rollout.

| Model | # Params | Train rel. $L_2$ | Test rel. $L_2$ | Rollout rel. $L_2$ |
|---|---|---|---|---|
| FNO XS | 1,642,261 | 0.014209 | 0.035924 | 0.806708 |
| FNO S | 4,203,937 | 0.008379 | 0.046619 | 1.044000 |
| FNO Base | 16,814,913 | 0.001404 | 0.051984 | 0.911689 |
| FNO L | 26,273,041 | 0.000999 | 0.048837 | 0.911812 |
| MSWT XS | 1,806,529 | 0.008744 | 0.030583 | 1.237877 |
| MSWT S | 3,947,404 | 0.005836 | 0.025568 | 0.752435 |
| MSWT Base | 13,531,924 | 0.002974 | 0.019826 | 0.347960 |
| MSWT L | 20,438,104 | 0.002716 | 0.017267 | 0.289241 |

### B.5.2. PARAMETER COUNT ABLATION

We compared the parameter efficiency of MSWT against FNO by varying the number of parameters with the two architectures and comparing their approximation errors. For the FNO ablation, we fixed the spectral resolution to $\texttt{modes}_x = \texttt{modes}_y = [16, 16, 16, 16]$ and varied the model capacity through the channel widths and projection size. Specifically, FNO-XS used $\texttt{layers} = [20, 20, 20, 20, 20, 20]$ with $\texttt{fc\_dim} = 40$, FNO-S used $\texttt{layers} = [32, 32, 32, 32, 32, 32]$ with $\texttt{fc\_dim} = 64$, FNO-Base used $\texttt{layers} = [64, 64, 64, 64, 64, 64]$ with $\texttt{fc\_dim} = 128$, and FNO-L used $\texttt{layers} = [80, 80, 80, 80, 80, 80]$ with $\texttt{fc\_dim} = 160$. For the MSWT ablation, we kept fixed $\texttt{patch\_size} = 2$, $\texttt{use\_efficient\_attention} = \text{True}$, $\texttt{efficient\_layers} = [0, 1]$, and $\texttt{local\_attention\_size} = 8$ and varied only the multi-scale channel dimensions. Specifically, MSWT-XS used $\texttt{dims} = [20, 20, 40, 160]$, MSWT-S used $\texttt{dims} = [32, 32, 64, 256]$, MSWT-Base used $\texttt{dims} = [64, 64, 128, 512]$, and MSWT-L used $\texttt{dims} = [80, 80, 160, 640]$.

The results of the parameter count ablation are shown in Table 10. The main highlight is that even a smaller version of our model (MSWT-S) with approximately 30% of the original parameter count is performing better than the baseline FNO model and its larger versions at test time. Hence, one cannot explain the differences in performance across the two architectures by parameter count alone. This, therefore, further indicates that the multi-scale wavelet-based architecture we propose is equipping the model with helpful inductive biases to capture small-scale phenomena, which is often missed by FNO-based architectures, but of special importance in chaotic dynamical systems.

### B.6. Runtime comparisons

We report average execution times for training and inference on the CKF benchmark in Table 11 and Table 12, respectively. Total training times were allocated so that each method was allowed a similar maximum amount of computational resources to be trained. The results show that MSWT runs at a similar time per training epoch to most other methods (except for SAOT, which takes much longer), whereas inference time is comparable to other wavelet-based architectures (WNO and SAOT). All our experiments were run on machines equipped with H100 NVIDIA GPUs and Intel Xeon Platinum 8452Y 36-core 2GHz processors on a high-performance computing cluster.

*Table 11.* Training time statistics (5 runs).

| Model | Total training time | Avg. per run | Avg. epochs/run | Avg. time/epoch (s) |
|---|---|---|---|---|
| FNO | 4h 40m 38.8s | 56m 7.8s | 100001.0 | 0.0337 |
| UNet | 5h 33m 36.3s | 1h 6m 43.3s | 100000.0 | 0.0400 |
| WNO | 7h 29m 12.2s | 1h 29m 50.4s | 64553.2 | 0.0835 |
| SAOT | 7h 29m 34.6s | 1h 29m 54.9s | 6170.0 | 0.8744 |
| HFS | 5h 7m 21.6s | 1h 1m 28.3s | 100001.0 | 0.0369 |
| MSWT | 6h 20m 22.4s | 1h 16m 4.5s | 100000.0 | 0.0456 |

*Table 12.* Inference timings (averaged over 100 samples).

| | FNO | UNet | WNO | SAOT | HFS | MSWT |
|---|---|---|---|---|---|---|
| Sec/trajectory | 0.1676 | 0.2653 | 0.6868 | 0.7246 | 0.2192 | 0.7449 |
| ms/step | 2.619 | 4.145 | 10.731 | 11.322 | 3.425 | 11.639 |

*Table 13.* Training runtime and long-horizon prediction error on CKF, illustrating the accuracy–runtime trade-off for FNO and MSWT.

| Model | Avg. time per training run | Long-horizon prediction error (step 64) |
|---|---|---|
| FNO | 56 min | $0.91 \pm 0.02$ |
| MSWT | 76 min | $0.33 \pm 0.03$ |

While FNO is the fastest model in wall-clock time, MSWT reduces long-horizon error by approximately 64% ($0.91 \rightarrow 0.33$) at a 36% increase in runtime, demonstrating a substantially improved accuracy–runtime trade-off for long-horizon prediction, as shown in Table 13. Figure 2 in the main paper shows that at this stage FNO is completely inaccurate and unstable across all wavenumbers, including the most energetic low wavenumbers (see Figure 2b), while our method still tracks the ground-truth spectrum accurately even after 64 steps of autoregressive rollout. Hence, comparison in terms of runtime efficiency alone is not a meaningful measure of performance for these applications. Instead, our results are essentially showing that MSWT achieves a significantly improved long-horizon predictive performance while maintaining training costs comparable to other wavelet-based methods (WNO and SAOT). We also note that further runtime improvements can be attained by code optimization, which was not our focus for this paper but remains as a potential direction for future work.

