# OpenReview forum: "Multi-Scale Wavelet Transformers for Operator Learning of Dynamical Systems"
_ICML.cc/2026/Conference — ICML 2026 regular_

### Official Review · Reviewer_gGv5 · 2026-03-11

**Soundness:** 2
**Presentation:** 3
**Significance:** 2
**Originality:** 2
**Overall Recommendation:** 3
**Confidence:** 3

**Summary:**

This paper proposes MSWT, a multi-scale wavelet transformer for operator learning in dynamical systems. The basic idea is to combine patch tokenization, wavelet-based down/up-sampling, and wavelet space attention so that high-frequency information is preserved more explicitly than in standard neural operators. The paper evaluates on CKF, SWE, and ERA5, and shows better performance on long-horizon prediction tasks.

**Compliance With Llm Reviewing Policy:**

Affirmed.

**Final Justification:**

It appears that FNO is faster in both training and inference, so the claim that MSWT is more efficient is not well supported. Therefore, I will maintain my score. I am willing to defer to the judgment of the other reviewers and the Area Chair if they believe the paper is strong enough to be accepted.

**Key Questions For Authors:**

(i) Can the authors provide a component ablation like wavelet attention vs. ordinary attention, wavelet-preserving sampling vs. standard sampling?

(ii) Can the authors report runtime or other efficiency metrics to better support the efficiency claim?

(iii) Can the authors include more baseline comparisons for the ERA5 weather prediction experiment? There appears to be a substantial body of prior work on neural network based weather forecasting, such as ClimODE and FourCastNet. I am not familiar with the weather prediction field, but comparing only against LUCIE seems somewhat limited and makes it harder to assess the value of the proposed method.

**Limitations:**

Yes.

**Strengths And Weaknesses:**

Strength:

(i) The problem is well motivated, and the proposed method is clearly presented.

(ii) The experimental results are comprehensive and provide support for some of the authors’ claims.

Weakness:

(i) It seems that there is already a lot of literature on using multiwavelet transforms in neural networks, so the novelty of the paper appears limited.

(ii) The paper does not adequately isolate why MSWT works. The paper claims the benefit comes from the representation and inductive bias, but the only explicit sensitivity analysis in the main paper is patch size. There is no clear component ablation for wavelet attention vs. ordinary attention, wavelet-preserving sampling vs. standard sampling, number of scales.

(iii) Efficiency claims are undersupported. The paper repeatedly argues that the method is more efficient because attention is performed in wavelet space and at reduced resolution, and even says it “reduces attention cost significantly.” But I did not see a direct runtime comparison in the main paper.

(iv) Several baselines become unstable and are reported as “–” on long rollouts.

(v) For the ERA5 weather prediction experiment, the paper appears to compare against only one baseline LUCIE, which makes the evaluation limited.

---

> ### Author Rebuttal · Authors · 2026-03-31
>
> We appreciate the reviewer's detailed feedback and important points. Regarding **ablations**, we refer the reviewer to our response to Reviewer `eWFZ`, while **runtime** details can be found in our response to Reviewer `pmmX`. We address the remaining concerns below.
>
> **MSWT** and **LUCIE** are both developed to emulate the long-term climate over multiple decades as opposed to the separate and distinct challenge of near-term weather prediction of lead times typically up to several weeks ahead. For example, **FourCastNet 3** (Bonev et al 2025) employs pretraining using 1024 NVIDIA H100 GPUs on 332,800 samples taking 78 hours followed by a subsequent second pre-training stage utilizing 512 NVIDIA A100 GPUs on 55,460 samples taking 15 hours to complete. A subsequent fine-tuning stage on 5840 samples requires 8 hours on 256 NVIDIA H100 GPUs.  FourCastNet 3 employs both local and global spherical convolution kernels to represent various scale dependent physical weather phenomena and a probabilistic objective function.
>
> In general, **weather** prediction systems are more concerned with uncertainty quantification and generating large ensembles and less concerned with longer-term model stability or computational expense. For example, the GNN-based **GraphCast** (Lam et al., 2023) with 36.7 million parameters takes 4 weeks on 32 Cloud TPU v4 devices using batch parallelism with performance at 10-day lead times displaying clear spectral bias at small scales across a range of standard diagnostics. Other GNN weather prediction variants such as **AIFS** (Lang et al., 2024a) are similar in performance to GraphCast
>
> Whereas it is simply not practical to train FourCastNet 3 without considerable computational resources, **LUCIE** nevertheless shares a common SFNO architecture (Bonev et al 2023) as FourCastNet 3 while exhibiting similar performance in terms of spectral bias and stability. Hence, it is a reasonable choice for comparison to MSWT, which is also able to mitigate spectral bias during very long autoregressive rollouts without recourse to increased complexity (such as the inclusion of denoising diffusion processes, etc).
>
> An additional constraint on long-range **climate** prediction is the lack of a sufficiently long historical record of the climate system. As the satellite record largely began only in the late 1970s, reliable training data is typically limited to the early 1980s onward. The climate system exhibits variability ranging from days to decades to centennial timescales. Hence, climate prediction is inherently a small-data challenge. Recent studies (O'Kane et al., 2026) demonstrated that variants of LUCIE display physical consistency with the observational satellite record without over smoothing in multi-decadal inference studies of tropical variability and radiative forcing ablation studies.
>
> We will make sure that the main aspects of these differences between weather and climate modeling are discussed in our revision of the paper.
>
> ### References
>
> - Bonev, B., Kurth, T., Hundt, C., Pathak, J., Baust, M., Kashinath, K., & Anandkumar, A. (2023). **Spherical Fourier neural operators: Learning stable dynamics on the sphere**. *Proceedings of the 40th International Conference on Machine Learning*, 202, 2806-2823.
> - Bonev, B., Kurth, T., Mahesh, A., Bisson, M., Kossaifi, J., Kashinath, K.,  ..., & Keller, A. (2025). **FourCastNet 3: A geometric approach to probabilistic machine-learning weather forecasting at scale**. *arXiv preprint arXiv:2507.12144*.
> - Lam, R. *et al.* (2023). **Learning skillful medium-range global weather forecasting**. *Science*, 382, 1416-1421.
> - Lang, S., Alexe, M., Chantry, M., Dramsch, J., Pinault, F., Raoult, B.,  ..., & Rabier, F. (2024). **AIFS-ECMWF's data-driven forecasting system**. *arXiv preprint arXiv:2406.01465*.
> - O'Kane, T. J., Collier, M. A., Sun, Y., Kitsios, V., & Arcomano, T. (2026). **On the role of radiative forcing in inference of tropical convection using a neural operator climate emulator**. https://doi.org/10.22541/essoar.177024078.89023797/v1

---

> > ### Author Rebuttal · Reviewer_gGv5 · 2026-04-04
> >
> > I have no further questions. It appears that FNO is faster in both training and inference, so the claim that MSWT is more efficient is not well supported. Therefore, I will maintain my score. I am willing to defer to the judgment of the other reviewers and the Area Chair if they believe the paper is strong enough to be accepted.

---

> > > ### Author Response · Authors · 2026-04-08
> > >
> > > We thank the reviewer for this observation. We agree that FNO achieves lower absolute runtime due to its simpler architecture. Our intended claim is not that MSWT is universally faster, but that it achieves improved long-horizon predictive performance while maintaining training costs comparable to standard baselines and competitive efficiency relative to other wavelet-based methods (WNO and SAOT). We will revise the wording in the paper to reflect this distinction more precisely.
> > >
> > > We agree that the term "efficiency" can be interpreted as referring to wall-clock training and inference time relative to FNO. What we intended by efficiency is, instead, **efficiency with respect to model capacity and long-horizon predictive performance**. In particular, our post-submission parameter-count ablation (provided in response to Reviewer pmmX) shows that MSWT achieves substantially lower rollout error with fewer parameters than FNO (e.g., MSWT-Base vs. FNO-Base), and that even smaller MSWT variants outperform larger FNO models at test time. Thus, for a fixed capacity budget, MSWT attains better long-horizon accuracy, and conversely, to reach a given rollout error it requires less model capacity. We believe this is the more relevant notion of efficiency for chaotic dynamical systems, where the primary objective is stable long-horizon prediction rather than raw throughput alone.
> > >
> > > To make this trade-off explicit, we include the following table comparing runtime and long-horizon error (at time step 64) for FNO and MSWT on the chaotic Kolmogorov flow (CKF) benchmark found in our experiments. While FNO is faster in wall-clock time, MSWT reduces long-horizon error by approximately 64% (0.91 → 0.33) at a 36% increase in runtime, demonstrating a substantially improved accuracy–runtime trade-off for long-horizon prediction. Figure 2 in our paper shows that at this stage FNO is completely inaccurate and unstable across all wavenumbers, even the most energetic low wavenumbers (see Figure 2b), while our method still tracks the ground-truth spectrum accurately even after 64 steps of autoregressive rollout. Hence, comparison in terms of runtime efficiency is not a meaningful measure of performance for these applications.
> > >
> > > > Training **runtime** and **long-horizon prediction error** (step 64) on CKF, illustrating the accuracy-runtime trade-off between FNO and MSWT
> > > | Model | Avg. time per run | long-horizon error (step 64) |
> > > |---|---:|---:|
> > > | FNO | 56 min | 0.91 ± 0.02 |
> > > | MSWT | 76 min | 0.33 ± 0.03 |
> > >
> > > We agree, however, that the manuscript should distinguish these notions of efficiency more carefully. In the revision we will:
> > >
> > > 1. replace generic claims of “improved efficiency” with more precise statements such as “reduced attention complexity” or “improved parameter efficiency”;
> > > 2. explicitly separate asymptotic/architectural efficiency from end-to-end wall-clock runtime; and
> > > 3. include timing results so that readers can directly assess the runtime–accuracy trade-off.
> > >
> > > We hope this clarifies that the central contribution of our work is improved long-horizon stability and accuracy, with efficiency understood in the sense above rather than as a claim of faster raw runtime than FNO.

---

### Official Review · Reviewer_eWFZ · 2026-03-11

**Soundness:** 3
**Presentation:** 4
**Significance:** 3
**Originality:** 2
**Overall Recommendation:** 5
**Confidence:** 3

**Summary:**

The authors address key challenges in Neural Operator Learning by presenting a Multi-Scale Wavelet Transformer for modeling dynamical systems. This approach, combining the wavelet transform with a Transformer architecture, outperforms prior methods on benchmarks such as the Chaotic Kolmogorov Flow, Shallow Waters, and ERA5 Climate analysis. The results highlight the model’s potential to advance the field.

**Compliance With Llm Reviewing Policy:**

Affirmed.

**Final Justification:**

The author fully addressed my concerns in the rebuttal. In particular, they were convincing in the ablation study and properly showed training/inference time, as requested. I especially appreciated the ablation study showing the effects of the modules. I recommend this paper to the conference.

**Key Questions For Authors:**

1. What are the training times and inference times for your method and in comparison to other methods? What hardware did you use?
2. What is the impact of each architectural choice on the performance of the method? Especially the patchify method, and the DWT?
3. Why did the other wavelet-based methods have significantly worse results?

**Limitations:**

yes

**Strengths And Weaknesses:**

Strengths:
1. Well-written paper, and clear explanation
2. Results are really impressive for the benchmarks shown, and the results are convincing for most of the analyses.
3. The claim and idea of using a wavelet transform to preserve high-frequency content is a promising idea.

Weaknesses:
1. The methodological analysis lacks depth and detail, making it difficult to distinguish the unique contributions of this work compared to important references such as FNO[4]. This limits readers’ ability to fully understand what sets this approach apart.
2. The absence of component-level ablation studies means there is no clear identification of which architectural features directly drive the observed performance gains.
3. Some claims, such as improved stability and spectral bias (line 191), lack either empirical demonstrations or theoretical justification, reducing the trustworthiness of these assertions.
4. It is unclear to what extent the wavelet transform alone, as opposed to its integration with the transformer, contributes to improvements, due to the lack of isolation and analysis of this component.
5. As baselines like WNO [1] and SAOT [3] also use wavelets, the source of improvement over these methods is not clearly demonstrated. SAOT also uses Transformers, so the authors should explain the main difference more clearly. An ablation study would clear these doubts.
6. Without clear component quantification, claims about wavelet attention capturing high-frequency patterns are speculative.
7. The paper does not provide details on training and inference times, despite noting that the architecture seems more complex than FNO. It is unclear whether this additional complexity results in longer training durations, and details on the hardware used are also missing.
8. Key prior work, such as “Multiwavelet-based operator learning for differential equations” [2], is not cited.


References:
[1] Tripura, T. and Chakraborty, S. Wavelet neural operator for solving parametric partial differential equations in computational mechanics problems. Computer Methods in Applied Mechanics and Engineering, 404:115783, 2023.
[2] Gupta, Gaurav, Xiongye Xiao, and Paul Bogdan. "Multiwavelet-based operator learning for differential equations." Advances in neural information processing systems 34 (2021): 24048-24062.
[3] Zhou, C., Chen, J., and Yang, Z. SAOT: An enhanced locality-aware spectral transformer for solving PDEs. In AAAI, 2026.
[4] Li, Z., Kovachki, N. B., Azizzadenesheli, K., Bhattacharya, K., Stuart, A., Anandkumar, A., et al. Fourier neural operator for parametric partial differential equations. In International Conference on Learning Representations, 2021.

---

> ### Author Rebuttal · Authors · 2026-03-31
>
> We thank the reviewer for their carefully detailed feedback and comments. We answer the questions below.
>
> **Hardware.** All our experiments were run on machines equipped with **NVIDIA H100 GPUs** and **Intel Xeon Platinum 8452Y 36-core 2 GHz** processors on a high-performance computing cluster. Training and inference times for ours and other methods are available in our response to Reviewer `pmmX`.
>
> **Ablation.** Regarding the impact of each architectural choice, we investigate the effectiveness of the three main modules of our framework separately,
>
> 1. **patch tokenizer**,
> 2. **wavelet attention block**, and
> 3. **wavelet-preserving down-/up-sampling**,
>
> in the following three variants of the model for ablation studies:
>
> - **MSWT-V1 (no tokenizer)** uses patch size of 1 for the patch tokenizer to learn the point-wise mapping of each point on the grid;
> - **MSWT-V2 (no attention)** removes the wavelet attention block and only keeps the UNet-based architecture with wavelet-preserving down-/up-sampling modules;
> - **MSWT-V3 (Conv-downsampling)** replaces the wavelet-preserving down-/up-sampling modules with standard convolutions/deconvolutions with a stride of 2 and keeps the wavelet attention blocks.
>
> We validate these three variants on the CKF benchmark and show the results in the following table. Removing the patch tokenizer (V1) improves the performance at the expense of efficiency, whereas removing the attention modules (V2) significantly damaged the performance, manifesting the effectiveness of the wavelet attention block. Using the standard strided-convolution/deconvolution (V3) also achieved sub-optimal results, showing the effectiveness of our proposed wavelet-preserving downsampling and upsampling modules. These ablation results will be included in our revised paper.
>
> > **ARCHITECTURE ABLATION**
> | Model | Train time | step 1 (Rel $L^2$) | step 30 | step 64 | step 1 (EMLR) | step 30 | step 64 |
> |-------|------------|-------------------|---------|----------|----------------|----------|----------|
> | **MSWT** | 1h 17' | 0.02 | 0.21 | 0.35 | 0.02 | 0.15 | 0.20 |
> | **MSWT-V1 (no tokenizer)** | 3h 4' | **0.01** | **0.10** | **0.18** | **0.01** | **0.07** | **0.10** |
> | **MSWT-V2 (no attention)** | 35' | 0.06 | 0.81 | 1.09 | 0.04 | 0.69 | 1.32 |
> | **MSWT-V3 (Conv-downsampling)** | 1h 4' | 0.03 | 0.51 | 0.71 | 0.03 | 0.32 | 0.39 |
>
> **Other wavelet-based methods.** There are a few significant differences between MSWT and other wavelet-based methods.
>
> - Both **WNO** and **SAOT** operate on full-resolution inputs. Without patch tokenization, both models need to learn representations over a much higher-dimensional domain, which often in practice requires much more data and modeling capacity. MSWT, instead, represents physical fields in token space, allowing us to compress the representation of the dynamics.
> - **WNO** has a similar architecture to FNO, replacing the FFT with a DWT (Daubechies wavelets), but still performs a *linear* convolution between the transform and its inverse. We instead use **attention**, a non-linear operation that can better capture interactions across different parts of the input.
> - **SAOT** has a more complex architecture involving both (Haar) wavelet and Fourier attention followed by a gated fusion block. While it is hard to pinpoint exactly what leads to its worse performance, we suspect that the extra complexities introduced by the architecture greatly increase the difficulty of learning optimal representations with such model. Note, for instance, the much longer amount of time per training epoch taken by SAOT when compared to other methods in the training times table in our response to Reviewer `pmmX`.
> - Neither WNO nor SAOT have a mechanism to handle domain **boundaries**, for example, the spherical domain for climate modeling, which we handle via **circular padding** in our patch tokenizer.
>
> We will add the missing related work suggestion [2] to our related work discussion in the revision, alongside the main points from the discussion above.

---

> > ### Author Rebuttal · Reviewer_eWFZ · 2026-04-02
> >
> > I appreciate the efforts the authors have made to address my concerns. The authors have addressed all my concerns, especially regarding the ablation study and training/inference time. I especially appreciated the ablation study showing the effects of the modules. Due to this, I will raise the score to an accept.

---

> > > ### Author Response · Authors · 2026-04-03
> > >
> > > Thank you for your timely response. We are glad our rebuttal and additional ablation studies addressed your concerns.
> > >
> > > We sincerely appreciate your recommendation for acceptance.

---

### Official Review · Reviewer_B7gv · 2026-03-13

**Soundness:** 2
**Presentation:** 3
**Significance:** 2
**Originality:** 2
**Overall Recommendation:** 4
**Confidence:** 4

**Summary:**

The authors propose Multi-Scale Wavelet Transformer (MSWT), a framework for mitigating the spectral bias via wavelet attention. Specifically, the leverage a U-Net alike architecture, to project the hidden features into multiple scales in a down and up-sampled fashion. Empirically, the conduct experiments on ERA5, and show a clear error reduction on short and relatively long-term step rollouts.

**Compliance With Llm Reviewing Policy:**

Affirmed.

**Final Justification:**

I read the authors' rebuttal in terms of training & testing time comparison. I am willing to increase my score due to the performance gain the proposed method achieves. However, the comparison still leaves me some concern: as (1) the inference time of the proposed method is five times larger than the baseline; and (2) the FNO's performance degraded when using a larger backbone, which makes me concerned about if the authors have done a good job in early stopping to prevent the model from a likely overfitting. Therefore, I decide to slightly increase my score, but leave the final decision to AC's judgement.

**Key Questions For Authors:**

1. What specific mechanism in the proposed architecture leads to the reported efficiency improvements? What's the empirical comparison regarding the unrolling time for different methods?

2. How do you handle the boundary?

3. How to extend to irregular meshgrids?

**Strengths And Weaknesses:**

### Strength

1. The paper is clearly written and easy to follow.

2. The spectral bias is an important question to address.

### Weakness

1. The authors talk about the improvements in efficiency as their main contribution. However, it's not clear the reason for this and the discussion in Section 3 does not address this fully.

2. The theoretical motivation for adopting wavelets as the multi-scale representation is not sufficiently justified. While wavelets are commonly believed to decompose signals into different frequency components, it is not clearly explained why the proposed formulation would specifically improve long-range dependency modeling or spectral representation.

3. What's the performance for unrolling the method to a very long period?

---

> ### Author Rebuttal · Authors · 2026-03-31
>
> We would like to thank the reviewer for their insightful comments and feedback. Regarding inference times, we refer the reviewer to our response to Reviewer `pmmX`. We address the remaining questions below.
>
> Our motivation for adopting wavelets is both **physical** and **architectural**. In many dynamical systems of interest, especially turbulent and geophysical flows, the solution contains both large-scale coherent structures and spatially localized fine-scale features (e.g., sharp gradients, shear layers, etc). Misrepresentation of this fine-scale content is particularly damaging over *long* rollouts of (quasi) two-dimensional flows, because the dynamics support both a forward enstrophy cascade and an inverse energy cascade (Kraichnan, 1967; Batchelor, 1969). Specifically, if the fine-scale content is oversmoothed/misrepresented, this perturbs the enstrophy flux and associated triadic interactions, with these errors propagating to larger and larger scales over autoregressive rollouts, ultimately leading to contamination of the large-scale structures. This is consistent with our results, where failure by the baseline models to capture small scales at earlier steps is later reflected in distorted spectra at low wavenumbers.
>
> In our MSWT, the discrete wavelet transform (DWT) provides an exact multiresolution decomposition into one low-pass (*approximation*) subband and three high-pass (*detail*) subbands, while retaining spatial localization. For the Haar basis, the approximation subband corresponds to local averages and the detail subbands to local differences in different directions. So the representation explicitly separates coarse structure from local detail. This means that coarse and fine scales are both represented explicitly, rather than asking the network to recover high-frequency content implicitly from a representation dominated by low-frequency energy (e.g., Fourier modes).
>
> The benefit for *long-range* dependency modeling comes from the combination of this representation with the architecture. Our MSWT applies attention after the DWT in a reduced-resolution wavelet space in which all subbands are retained. This makes distant interactions cheaper to model, while the multi-scale hierarchy captures broad interactions on coarse levels and the inverse DWT together with skip connections restores local detail at fine levels. The key novelty is therefore **not wavelets alone**, but **wavelets plus attention plus wavelet-preserving down/up-sampling**. Together, they allow global/coarse interactions to be modeled without discarding the fine-scale content that is often responsible for long-horizon instability. This is also consistent with our empirical results, where MSWT yields substantially improved spectral metrics and long-horizon rollout stability compared to the baseline models.
>
> **Irregular meshes.** Our current implementation does assume a regular grid and does not directly handle arbitrary unstructured meshes: the formulation is written for discretized fields on a $H\times W$ lattice and uses a separable 2D Haar DWT. However, while out of scope for the current paper, the architecture may be extended to unstructured meshes in a fairly natural way. The essential ingredients of MSWT are not tied to Cartesian grids per se, what is really needed is:
>
> 1. an invertible multi-resolution decomposition,
> 2. explicit approximation/detail channels,
> 3. attention applied in the corresponding reduced-resolution coefficient space.
>
> Some examples of how this might be achieved include replacing the regular-grid DWT with a lifting-based biorthogonal transform on the unstructured mesh, or replacing it with a graph/manifold wavelet transform built from the mesh Laplacian. An alternative route would be to retain the regular-grid DWT but replace the patch tokenizer/inverse tokenizer with a geometry-aware encoder/decoder that maps between an arbitrary point cloud and a regular latent grid.
>
> **Boundaries.** In our current implementation, we adopt circular padding (with patch size greater than 1) to handle boundaries on spherical domains, but an interesting direction for future work would be to investigate alternative approaches for enforcing arbitrary boundary conditions on general domains.
>
> **Unrolling for a long period.** SWE predictions were rolled out for 80 steps, and the ERA5 benchmark was rolled out for 6-hourly data for 10 years.
>
> ### References
>
> * Kraichnan, R. H. (1967). *Inertial Ranges in Two-Dimensional Turbulence*. *Physics of Fluids*, 10, 1417 --1423.
> * Batchelor, G. K. (1969). *Computation of the Energy Spectrum in Homogeneous Two-Dimensional Turbulence*. *Physics of Fluids*, 12, II-233-II-239.

---

> > ### Author Rebuttal · Reviewer_B7gv · 2026-04-03
> >
> > I still hold concern (1) regarding the improvement of long-term predictions, since the multi-scale transform is performed on local state variable, and thus not sufficient to show why this temporal forecasts; (2) the inference time is rather an ablation instead of a comparison in terms of different architectures. Therefore I decide to keep my score.

---

> > > ### Author Response · Authors · 2026-04-08
> > >
> > > * * *
> > >
> > > We appreciate the reviewer’s acknowledgment of our previous rebuttal and for detailing their remaining concerns, which we summarize and address next.
> > >
> > > 1. **Why can a wavelet-based method, which typically performs well on modeling spatial scales, demonstrate superior performance in long-term temporal rollouts?**
> > >
> > >    **Response.** We refer the reviewer to Fig. 6, Fig. 8, and Fig. 10 in the appendix of our paper, which illustrate the key mechanism underlying the performance improvements of our method. These figures present spectral representations of multiple methods at different time steps. **They particularly show that low-frequency modes are captured well by most methods, and the primary source of long-horizon degradation is the accumulation of high-frequency errors over rollout.**
> > >
> > >    This behavior is consistent with the discussion in our paper on spectral bias and long-horizon instability. In nonlinear multiscale systems, small-scale features are not dynamically negligible: errors at high frequencies (small spatial scales) perturb inter-scale energy transfer (e.g., enstrophy flux), and these perturbations propagate to larger scales over time. As a result, inaccuracies at early steps lead to progressively distorted large-scale structure during the autoregressive rollout over time.
> > >
> > >    The role of the DWT in MSWT should be understood in this context. As described in Sec. 4, the DWT is a local multiresolution change of basis that explicitly separates coarse and fine scales. It is not itself a mechanism for long-range temporal modeling. The improvement arises from its combination with the architecture: attention is applied in reduced-resolution wavelet spaces, while all subbands are preserved and mixed across scales. This allows MSWT to model long-range spatial interactions efficiently without discarding fine-scale content.
> > >
> > >    Crucially, this design avoids the systematic misrepresentation of high-frequency (fine-scale) components at each time step. Since such errors would otherwise accumulate under autoregressive dynamics, their mitigation directly improves long-term temporal stability. This is precisely what is observed in our spectral results: MSWT better preserves high-frequency energy, exhibits reduced spectral distortion, and achieves more stable long-horizon rollouts than the baselines.
> > >
> > > 2. **the inference time is rather an ablation instead of a comparison in terms of different architectures**
> > >
> > >    **Response.** We refer the reviewer to our response to Reviewer `pmmX`, where we report separate tables for training and inference times (**TRAINING TIMES** and **INFERENCE TIMES**). These tables provide direct comparisons across different architectures (FNO, U-Net, WNO, SAOT, HFS, and MSWT), rather than ablations over a single model. The key observation is that MSWT achieves training costs comparable to the state-of-the-art method HFS, which relies primarily on simple convolutional operations. In contrast, MSWT incorporates more complex attention mechanisms, yet still maintains similar training time, while producing significantly lower long-horizon prediction errors as shown in our experimental results. Furthermore, MSWT demonstrates clear computational efficiency advantages over other wavelet-based architectures, such as WNO and SAOT, which incorporate similar components, but incur higher computational cost (WNO and SAOT take roughly 2x and 20x more time per training epoch, respectively, than MSWT). These results suggest that MSWT not only improves predictive performance but does so without introducing additional computational overhead relative to other wavelet-based architectures, while keeping a similar average training time across other baselines (see average time per epoch).
> > >
> > > If our clarifications regarding temporal predictive performance and computational efficiency address the reviewer’s concerns, we would greatly appreciate a reconsideration of the review score.

---

### Official Review · Reviewer_pmmX · 2026-03-13

**Soundness:** 3
**Presentation:** 3
**Significance:** 3
**Originality:** 3
**Overall Recommendation:** 4
**Confidence:** 4

**Summary:**

This paper proposes a method, multi-scale wavelet transformers (MSWT) that mitigates the spectral bias problem of neural operators by utilizing a wavelet-preserving downsampling scheme and wavelet-based attention across scales and frequency bands. The spectral bias of neural operators is a well-known problem characterized by the failure to learn high-frequency components. This problem has motivated many previous works in this area; the authors provide a comprehensive review of the existing methods and contextualize their method within this literature. The authors provide experiments against baselines FNO, U-Net, WNO, SAOT, and HFS on the 2D Kolmogorov flow and the shallow water equation; and a baseline (LUCIE) on the ERA5 climate reanalysis dataset, showing superior performance on rollouts.

**Compliance With Llm Reviewing Policy:**

Affirmed.

**Final Justification:**

After great thought, I have decided to maintain my score of weak accept. I applaud the authors for sharing their code to resolve the issue of the size of their model; I am confident in the claim by the authors. However, I share the concerns of reviewers B7gv and gGv5 regarding the strength of the baselines (whether they are matched), and the lower efficiency of the method i.e. much longer inference time, even up to half an order of magnitude. I believe these are very important points when considering the practical usefulness of the method. I thus entrust the final decision to the AC.

**Key Questions For Authors:**

**Questions**
- 1) *Full code availability*: I saw that the supplementary material is a link to the anonymized code repo. The code seems to be very minimal, containing only a minimal implementation of the method but without any of the code that would reproduce the results presented in the paper. Would the authors be willing to open-source the full codebase after the anonymity period?
- 2) In relation to *Weakness 1* I am curious about the parameter efficiency of the MSWT in the sense of how does the number of additional parameters, compared to an FNO baseline, scale with increasing model size?
- 3) *Training details*: Could the authors provide more details on the training and inference speed/costs for MSWT and each of the baselines?

**Limitations:**

**Limitations**:
See **Questions** section.

**Strengths And Weaknesses:**

**Strengths**
- 1) *Contextualization of the work within the existing literature*: The authors did a good job in describing the problem of the spectral bias of neural operators, and in introducing two aspects i.e. architectural bottlenecks such as Fourier-mode truncation as well as MSE-type training objects that over-smooth. They also were comprehensive in explaining the shortcomings and positioning of existing work: 1) existing wavelet-based neural operators rely on less-expressive CNNs; 2) existing transformer neural operator hybrids apply attention on the full-resolution grid, which incurs a high cost; and 3) modified objective functions and rollout regularization are orthogonal to architectural innovations and diffusion-based neural operator methods (of where there are several) have an unavoidable inference-time overhead. The proposed MSWT can be seen as a refinement of ideas from both 1) and 2), and I also applaud the authors' choice to keep the MSE-type loss fixed, to isolate the role of representation/inductive bias (wavelets and the downsampling).
- 2) *Clarity of the methods section*: I believe the authors employed the appropriate level of formality in their methods section, and I appreciate the amount of detail provided. It was readable and informative, with some insightful explanations and discussion of tradeoffs. There is sufficient additional detail in the Appendix, which was useful for some preliminary knowledge.
- 3) *Well-motivated method*: The MSWT method appears to be well-motivated, as a wavelet-based method has many appealing properties. The 2D Haar DWT seems to be a natural choice. The authors also explained the spectral bias problem well, discussing several lines of prior work that have investigated it.
- 4) *Transparent and detailed evaluation setting*: I appreciate the plentiful details on the evaluation setting presented in the main text and appendix. I also appreciate the study on the patch size, and I think it's not at all surprising that increasing patch size after a certain point results in worse performance. The presentation of the enstrophy power spectrum for the Kolmogorov flow is also well-motivated.

**Weaknesses**
- 1) *Parameters of baselines aren't as well-matched as claimed*: Judging from Table 4 in the Appendix (which I applaud the authors for including) it seems that the MSWT still has significantly more parameters (19.3M compared to e.g. 12.5M for U-Net, 14.5M for WNO, and 16.8M for FNO, as some representative examples). While I understand that matching the parameter count is challenging, it seems that the gap is significant. Could the authors comment on this point, as well as explain the choices of the $n_{\text{hidden}}$ values for each model presented in Table 4? I will strongly consider increasing my score upon a satisfactory explanation, and apologies in advance if I have misinterpreted the presentation of these baselines.

---

> ### Author Rebuttal · Authors · 2026-03-31
>
> We would like to thank the reviewer for their careful consideration of our paper and detailed feedback. We detail our response below.
>
> **Code release.** Yes, we are planning to open source the full codebase upon acceptance.
>
> **Parameter count.** Post-submission we realized that the parameter count in Table 4 was for a previous version of MSWT, not the one that we used for experiments. The version we used for experiments had a parameter count of **13M**, which is actually lower than FNO's count. All models were configured to have a similar capacity to that of FNO considering the numbers of layers and hidden units, with remaining hyper-parameters set to ensure a similar parameter count, which is hard to exactly match given the very different architectures.
> Note also that an exact match in parameter count does not guarantee the same modeling capacity across different architectures (e.g., a purely linear model would never be able to represent a quadratic function, which a polynomial model might easily do).
> However, considering the reviewer's suggestion, we compared the parameter efficiency of MSWT against FNO by varying the number of parameter of the two architectures and comparing their approximation errors.
>
> The results of the parameter count ablation are shown in the table below. We defer the details of the ablation to further below, but the main insight is that even a smaller version of our model (MSWT-S) with approximately 30\% of the original parameter count is doing better than the baseline FNO model and its larger versions at test time. Hence, one cannot explain the differences in performance across the two architectures by parameter count alone. This, therefore, further indicates that the multiscale wavelet-based architecture we propose is equipping the model with helpful inductive biases to capture small scale phenomena, which is often missed by FNO-based architectures, but of special importance in chaotic dynamical systems.
>
> >  **PARAMETER COUNT ABLATION**
> | Model | # Params | Train rel. $L_2$ | Test rel. $L_2$ | Rollout rel. $L_2$ |
> |-------|----------|------------------|----------------|-------------------|
> | FNO XS | 1,642,261 | 0.014209 | 0.035924 | 0.806708 |
> | FNO S | 4,203,937 | 0.008379 | 0.046619 | 1.044000 |
> | FNO Base | 16,814,913 | 0.001404 | 0.051984 | 0.911689 |
> | FNO L | 26,273,041 | 0.000999 | 0.048837 | 0.911812 |
> | **MSWT XS** | 1,806,529 | 0.008744 | 0.030583 | 1.237877 |
> | **MSWT S** | 3,947,404 | 0.005836 | 0.025568 | 0.752435 |
> | **MSWT Base** | 13,531,924 | 0.002974 | 0.019826 | 0.347960 |
> | **MSWT L** | 20,438,104 | 0.002716 | 0.017267 | 0.289241 |
>
> **Training and inference times.** We report average execution times for training and inference on the CKF benchmark in the following tables. Total training times were allocated so that each method was allowed a similar maximum amount of computational resources to be trained. The results show that MSWT runs at a similar time per training epoch to most other methods (except for SAOT, which takes much longer), whereas inference time is mainly comparable to other wavelet-based architectures (WNO and SAOT). We, however, note that this consists of the first iteration of our method and further optimizations can later be done to optimize inference time. Our main focus on this paper was on reducing tracking errors over long rollouts.
>
> > **TRAINING TIMES** (averaged over 5 runs)
> | Model | Total training time | Avg. per run | Avg. epochs/run | Avg. time/epoch (s) |
> |-------|---------------------|--------------|-----------------|---------------------|
> | FNO | 4h 40m 38.8s | 56m 7.8s | 100001.0 | 0.0337 |
> | UNet | 5h 33m 36.3s | 1h 6m 43.3s | 100000.0 | 0.0400 |
> | WNO | 7h 29m 12.2s | 1h 29m 50.4s | 64553.2 | 0.0835 |
> | SAOT | 7h 29m 34.6s | 1h 29m 54.9s | 6170.0 | 0.8744 |
> | HFS | 5h 7m 21.6s | 1h 1m 28.3s | 100001.0 | 0.0369 |
> | **MSWT** | 6h 20m 22.4s | 1h 16m 4.5s | 100000.0 | 0.0456 |
>
> > **INFERENCE TIMES** (averaged over 100 samples)
> | Model | Sec/trajectory | ms/step |
> |-------|----------------|---------|
> | FNO | 0.1676 | 2.619 |
> | UNet | 0.2653 | 4.145 |
> | WNO | 0.6868 | 10.731 |
> | SAOT | 0.7246 | 11.322 |
> | HFS | 0.2192 | 3.425 |
> | **MSWT** | 0.7449 | 11.639 |
>
> **Parameter count ablation details.** For the FNO ablation, we fixed the spectral resolution to `modes_x = modes_y = [16,16,16,16]` and varied the model capacity through the channel widths and projection size. Specifically, we set:
>
> - **FNO-XS** : `layers = [20,20,20,20,20,20]` with `fc_dim = 40`,
> - **FNO-S** : `layers = [32,32,32,32,32,32]` with `fc_dim = 64`,
> - **FNO-Base** : `layers = [64,64,64,64,64,64]` with `fc_dim = 128`,
> - **FNO-L** : `layers = [80,80,80,80,80,80]` with `fc_dim = 160`.
>
> For the MSWT ablation, we varied only the multiscale channel dimensions (all other settings kept fixed). Specifically,
>
> - **MSWT-XS** : `dims = [20,20,40,160]`,
> - **MSWT-S** : `dims = [32,32,64,256]`,
> - **MSWT-Base** : `dims = [64,64,128,512]`,
> - **MSWT-L** : `dims = [80,80,160,640]`.

---

> > ### Author Rebuttal · Reviewer_pmmX · 2026-04-02
> >
> > I thank the authors for a detailed rebuttal. In the absence of the full codebase, it is impossible for me to judge whether it is truly the case that the authors made a typo in their manuscript, significantly overestimating the size of their model. I am still uncertain about this, but I will give the authors the benefit of the doubt.
> >
> > I also thank the authors for their transparency in providing the training and inference times. In particular, this elucidates the significant inference-time slowdown of the proposed method, but I do also tend to agree with the authors that this sort of thing can be reduced, to some extent, with code-level optimizations.
> >
> > With all these considerations in mind, I will maintain my score, but I emphasize the crucial points discussed above.
> >
> > I urge the authors to eventually open source the full codebase for transparency and to ensure it can be built upon.

---

> > > ### Author Response · Authors · 2026-04-03
> > >
> > > We thank the reviewer for acknowledging our efforts toward transparency, particularly in reporting training and inference costs, and for suggesting that we open-source the codebase. We fully agree that making the implementation publicly available is important for ensuring reproducibility and facilitating future research.
> > >
> > > In response, **we have open-sourced our model along with the baseline methods at the following link**:
> > >
> > > https://anonymous.4open.science/r/multiscale_wavelet_transformer_anoymous_review-1BE8/readme.md
> > >
> > > The reviewer is very welcome to verify the implementation by training the model on synthetic data (e.g., by running train_operator_AR_rell2_2d.py). Additional details are provided in the README file. Upon execution, the script should report **“Total Trainable Params: 13,531,924”** for our MSWT model.
> > >
> > > We also plan to release the code for the additional benchmarks in the future to further support transparency and reproducibility.
> > >
> > > If these clarifications help address your concerns, we would greatly appreciate your consideration in updating the score to reflect these improvements.

---

### Decision · Program_Chairs · 2026-04-30

**Decision:**

Accept (regular)

**Comment:**

This paper proposes a Multi-Scale Wavelet Transformer (MSWT) in time-dependent problems to mitigate the spectral bias via wavelet-based attention operator coupled with the common U-net meta-architecture. Performance-wise, it is competitive and tested on real-life datasets such as ERA5. Several reviewers questioned in the rebuttal about the experiment setup (e.g., params count) and I suggest the authors address those in the revision.